# Aquaporin-4-dependent glymphatic solute transport in the rodent brain

Humberto Mestre[1†], Lauren M Hablitz[1†], Anna LR Xavier[2†], Weixi Feng[3†], Wenyan Zou[3†], Tinglin Pu[3†], Hiromu Monai[4,5†], Giridhar Murlidharan[6†], Ruth M Castellanos Rivera[6†], Matthew J Simon[7†], Martin M Pike[8†], Virginia Plá[1†], Ting Du[1†], Benjamin T Kress[1†], Xiaowen Wang[4], Benjamin A Plog[1], Alexander S Thrane[2,9], Iben Lundgaard[1,10,11], Yoichiro Abe[12], Masato Yasui[12], John H Thomas[13,14], Ming Xiao[3*], Hajime Hirase[4,15*], Aravind Asokan[6,16,17*], Jeffrey J Iliff[7,18*], Maiken Nedergaard[1,2*]

[1]Center for Translational Neuromedicine, University of Rochester Medical Center, Rochester, United States; [2]Center for Translational Neuromedicine, Faculty of Medical and Health Sciences, University of Copenhagen, Copenhagen, Denmark; [3]Jiangsu Province Key Laboratory of Neurodegeneration, Center for Global Health, Nanjing Medical University, Nanjing, China; [4]RIKEN Center for Brain Science, Wako, Japan; [5]Ochanomizu University, Tokyo, Japan; [6]Gene Therapy Center, The University of North Carolina, Chapel Hill, United States; [7]Department of Anesthesiology and Perioperative Medicine, Oregon Health and Science University, Portland, United States; [8]Advanced Imaging Research Center, Oregon Health and Science University, Portland, United States; [9]Department of Ophthalmology, Haukeland University Hospital, Bergen, Norway; [10]Department of Experimental Medical Science, Lund University, Lund, Sweden; [11]Wallenberg Center for Molecular Medicine, Lund University, Lund, Sweden; [12]Department of Pharmacology,School of Medicine, Keio University, Tokyo, Japan; [13]Department of Mechanical Engineering, University of Rochester, Rochester, United States; [14]Department of Physics and Astronomy, University of Rochester, Rochester, United States; [15]Brain and Body System Science Institute, Saitama University, Saitama, Japan; [16]Department of Molecular Genetics and Microbiology, Duke University School of Medicine, North Carolina, United States; [17]Department of Surgery, Duke University School of Medicine, Durham, United States; [18]Knight Cardiovascular Institute, Oregon Health and Science University, Portland, United States

*For correspondence:
mingx@njmu.edu.cn (MX);
hajime.hirase@riken.jp (HH);
aravind.asokan@duke.edu (AA);
iliffj@ohsu.edu (JJI);
maiken_nedergaard@urmc.
rochester.edu (MN)

†These authors contributed equally to this work

Competing interests: The authors declare that no competing interests exist.

**Abstract** The glymphatic system is a brain-wide clearance pathway; its impairment contributes to the accumulation of amyloid-β. Influx of cerebrospinal fluid (CSF) depends upon the expression and perivascular localization of the astroglial water channel aquaporin-4 (AQP4). Prompted by a recent failure to find an effect of *Aqp4* knock-out (KO) on CSF and interstitial fluid (ISF) tracer transport, five groups re-examined the importance of AQP4 in glymphatic transport. We concur that CSF influx is higher in wild-type mice than in four different *Aqp4* KO lines and in one line that lacks perivascular AQP4 (*Snta1* KO). Meta-analysis of all studies demonstrated a significant decrease in tracer transport in KO mice and rats compared to controls. Meta-regression indicated that anesthesia, age, and tracer delivery explain the opposing results. We also report that intrastriatal injections suppress glymphatic function. This validates the role of AQP4 and shows that glymphatic studies must avoid the use of invasive procedures.
DOI: https://doi.org/10.7554/eLife.40070.001

## Introduction

A brain-wide fluid transport pathway, known as the glymphatic system, supports the rapid exchange of cerebrospinal fluid (CSF) and interstitial fluid (ISF) along perivascular pathways (*Iliff et al., 2012*). The glymphatic system consists of three principal sequential anatomic segments: (i) CSF inflow along the perivascular spaces surrounding penetrating arteries, (ii) dispersion of CSF through the wider interstitium, and (iii) efflux of ISF along the large-caliber draining veins to re-enter the CSF within the ventricular and cisternal compartments (*Jessen et al., 2015*). Ultimately, interstitial solutes cleared to the CSF exit the brain through meningeal lymphatic vessels flanking the venous sinuses, along cranial and spinal nerve sheathes, and across the cribriform plate (*Louveau et al., 2015*; *Aspelund et al., 2015*). Astrocytic endfeet ensheath the cerebral vasculature, and the abundantly expressed astroglial water channel aquaporin-4 (AQP4) localizes primarily to the perivascular endfoot membrane domain abutting the basal lamina. As this anatomic arrangement provides a route for rapid water movement between the perivascular space and the glial syncytium, AQP4 has been proposed to support perivascular fluid and solute movement along the glymphatic system (*Nedergaard, 2013*).

Several groups have independently shown that the astrocytic AQP4 is essential for fast glymphatic transport. Iliff et al. demonstrated a significant suppression of both perivascular CSF tracer influx and interstitial mannitol and amyloid-β (Aβ) clearance in *Aqp4* knockout (KO) mice (*Iliff et al., 2012*). Subsequent work demonstrated that *Aqp4* gene deletion exacerbated glymphatic pathway dysfunction after traumatic brain injury (TBI) and promoted the development of neurofibrillary pathology and neurodegeneration in the post-traumatic brain (*Iliff et al., 2014*). Plog et al. similarly found that *Aqp4* KO mice exhibit slowed transport of interstitial solutes to the blood after TBI, reflected by a significant reduction in plasma biomarkers of TBI, including GFAP, neuron-specific enolase, and S100β (*Plog et al., 2015*). Xu et al. reported that deletion of *Aqp4* exacerbated Aβ plaque accumulation and cerebral amyloid angiopathy in the APP/PS1 murine model of Alzheimer's disease (*Xu et al., 2015*). Achariyar et al. found significantly reduced distribution of FITC-ApoE3, $^{125}$I-apoE2, $^{125}$I-apoE3, $^{125}$I-apoE4, as well as $^{14}$C-inulin in *Aqp4* KO mice following tracer injection to the CSF (*Achariyar et al., 2016*). Lundgaard et al. reported that glymphatic clearance of lactate was reduced in *Aqp4* KO mice (*Lundgaard, 2016*). Finally, Murlidharan et al. demonstrated that *Aqp4* KO mice exhibit significantly impaired clearance of adeno-associated viruses (AAV) infused into the ventricles, and concluded that glymphatic transport profoundly affects various aspects of AAV gene transfer in the CNS (*Murlidharan et al., 2016*). A recent MRI study showed the AQP4 facilitator, TGN-073, potentiated the transport of interstitial fluid from the glia limitans externa to pericapillary Virchow-Robin space (*Huber et al., 2018*).

The critical role of AQP4 in supporting perivascular CSF-ISF exchange was recently questioned in a report by *Smith et al. (2017)*, in which the authors failed to detect any reduction in CSF tracer influx into the brain parenchyma of *Aqp4* KO mice compared to wild-type (WT) controls. Because a key element of the glymphatic hypothesis is the role of astroglial water transport in supporting perivascular CSF-ISF exchange, we consider it critical to re-examine the role of AQP4 in this process, with an aim to resolve the discrepant reports. Using data generated from five independent laboratories, we have undertaken such a re-evaluation of the effects of *Aqp4* deletion on perivascular glymphatic exchange. Results of this analysis consistently confirm that *Aqp4* deletion impaired perivascular glymphatic flow relative to that in wild-type mice. Our conclusion is strengthened by the use of four independently generated *Aqp4* KO lines, including the line used by *Smith et al. (2017)*, as well as the α-syntrophin (*Snta1*) KO line, which lacks AQP4 perivascular localization despite normal expression levels (*Amiry-Moghaddam et al., 2003*).

*Smith et al. (2017)* also questioned the existence of tissue bulk flow based injecting tracers of varying molecular sizes into cortex or striatum. We questioned this approach based on the prior finding that traumatic brain injury is linked to an immediated and sustained reduction in CSF influx (*Iliff et al., 2014*). As expected, our analysis showed that insertion of glass pipettes into the brain markedly reduced brain-wide CSF tracer influx. Thus, interstitial fluid transport should not be studied following invasive procedures in the brain.

## Results

The study included data from five laboratories using four independently generated *Aqp4* KO lines and one *Snta1* KO mouse line (*Figure 1a–e*) (*Fan et al., 2005*; *Ikeshima-Kataoka et al., 2013*; *Ma et al., 1997*; *Thrane et al., 2011*; *Adams et al., 2000*). Immunohistochemistry done in parallel with the glymphatic experiments verified that AQP4 was indeed deleted in all the *Aqp4* KO mouse lines (*Figure 1a–c and e*). In the *Snta1* KO mice, immunofluorescence demonstrated that perivascular AQP4 polarization was absent in this line (*Figure 1d*).

### NMU: reduced influx of a fluorescent CSF tracer in *Aqp4* KO mice

We injected the fluorescent tracer Texas Red-dextran (3 kD, TRd3) intracisternally, as previously described (*Iliff et al., 2012*). We first compared overall penetration of TRd3 into the dorsal and ventral surfaces of the whole-brain between WT and *Aqp4* KO (*Aqp4*$^{-/-}$) mice using ex vivo near infrared fluorescence imaging. Quantification of the mean integrated optical density (MIOD) of TRd3 on the dorsal and ventral brain surface along anterior-posterior position of bregma showed a significant reduction in *Aqp4* KO mice compared with WT mice (*Figure 2a–c*). Interestingly, CSF tracer entry into the ventral brain was higher than into the dorsal brain. We also compared penetration of TRd3 into the brain on the serial coronal forebrain slides (+1.7 to −0.7 mm from bregma) between the two genotypes. The percentage area of whole-slice fluorescence was significantly reduced in *Aqp4* KO mice compared to WT controls (*Figure 2d,f*). Subregional quantification on the coronal section at the level of 0.5 mm anterior to the bregma showed that penetration of CSF tracer was high in the ventral and lateral brain surface of WT mice, but was comparable in the dorsal surface of the brain with *Aqp4* KO mice (*Figure 2g–h*), further supporting difference in brain regions of CSF tracer influx. Specifically, *Aqp4* KO markedly impaired the influx of TR-d3 into both the perivascular space and the brain parenchyma in the hypothalamus, one of the brain regions with the highest expression of AQP4 (*Fan et al., 2005*; *Nielsen et al., 1997*) (*Figure 2e*). Quantification of the intensity of TR-d3 as a function of the distance from the brain surface showed rapid decay of tracer with increasing distance from the ventral surface of the hypothalamus in *Aqp4* KO mice. The tracer was almost undetectable at 500 μm below the brain surface in both the perivascular space and in the brain parenchyma in *Aqp4* KO mice. By contrast, the TR-d3 fluorescence intensity remained at 90% of the pial surface intensity to a depth of 500 μm in the perivascular space as well as in the adjacent parenchyma in WT mice (*Figure 2i–j*). Taken together, these results replicate the previous finding that AQP4 facilitates the transfer of intracisternally injected TR-d3 from the CSF into the parenchyma (*Iliff et al., 2012*).

### RIKEN: histological enhancement of CSF tracer reveals compromised parenchymal tracer infiltration in *Aqp4* KO mice

The RIKEN and Keio group used another strain of *Aqp4* KO mice, in which the exon 1 of *Aqp4* is replaced with *eGFP*. In this experiment, biotinylated dextran amine (BDA, 0.5%, 70 kDa) was injected under deep ketamine-xylazine (KX) anesthesia (*Figure 3a*). Because *Aqp4* KO mice express eGFP, we used Alexa 594-conjugated streptavidin (SA) to visualize BDA. This approach has an advantage that BDA signals are amplified by the histological processing and clearly distinguished from intrinsic fluorescence (i.e. before histological processing). As a result, BDA distribution appeared less extensive in the cortex of *Aqp4* KO mice (*Figure 3b*). Quantification of SA-enhanced BDA signals in the cortical parenchyma showed depth-dependent profiles that decayed with cortical depth after the initial peak for both wild type and *Aqp4* KO mice. Notably, BDA signals in the cortex of *Aqp4* KO mice are of lower intensity and penetrated less into the parenchyma at an anterior-posterior position of bregma (*Figure 3c*). Next, we examined if the compromised BDA penetration in *Aqp4* KO mice is generally observed across the anterior-posterior extent of the cortex. Indeed, *Figure 3d* shows that BDA distribution in *Aqp4* KO cortex is broadly compromised.

### UNC: CSF tracer influx is decreased in *Aqp4* KO mice compared to background strain controls and wild-type mice

We assessed the reproducibility of the experiments performed in *Smith et al. (2017)* by using the same *Aqp4* KO mouse line but employing the original methodology reported by *Iliff et al. (2012)*. In order to determine if the background strains of the different *Aqp4* KO lines have an effect on CSF

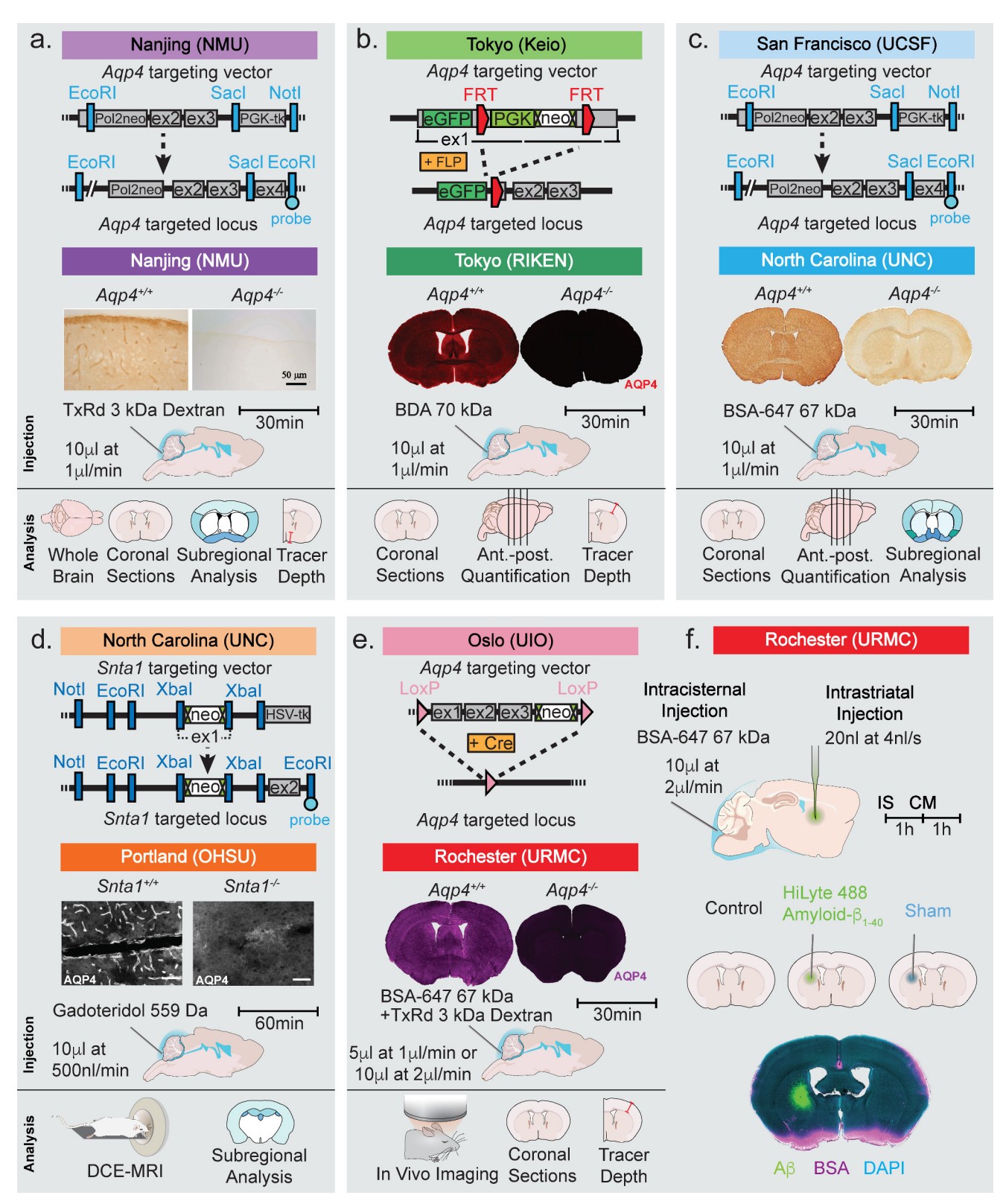

**Figure 1.** Strategy used for generation of KO mice and the experimental design of the study. Top row of each panel represents the institution where the line originated from and the strategy used to generate the four global *Aqp4* KO mice and the *Snta1* KO mice. Second row of each panel shows the five labs that collected data on glymphatic system function in the five transgenic mouse lines: (a) Nanjing Medical University (NMU), (b) RIKEN Center

*Figure 1 continued on next page*

Figure 1 continued

for Brain Science (RIKEN), (c) University of North Carolina (UNC), (d) Oregon Health and Science University (OHSU), (e) University of Rochester Medical Center (URMC). (a-e) Immunohistochemical analysis showing the lack of AQP4 expression in the global KOs and (d) lack of perivascular AQP4 localization in *Snta1* KO compared to WT mice. Scale bar: 50 µm. (a-e) Third row displays the volume and rate used for the intracisterna magna (CM) injections for each experiment, the tracer used, and the experiment duration. Note that URMC collected two full data sets using an injection rate of either 1 or 2 µl/min. (a-e) The last row display the analyses strategy employed by each of the five research group in *Figures 2–6*. (f) Additional experiments performed in *Figure 7* tested the effect of intrastrial (IS) injection on global glymphatic function. TxRd, Texas Red; BDA, biotinylated dextran amine; BSA-647, bovine serum albumin-Alexa Fluor 647; DCE-MRI, dynamic contrast-enhanced magnetic resonance imaging.

DOI: https://doi.org/10.7554/eLife.40070.002

inflow, we included both CD1 and C57BL/6 controls. We compared the intraparenchymal entry of Alexa Fluor 647-conjugated bovine serum albumin (66 kDa, BSA-647) delivered into cisterna magna of ketamine/xylazine-anesthetized mice. Thirty minutes after the injection, the brains were harvested, and tracer distribution evaluated in coronal sections as described previously (*Iliff et al., 2012*). Parenchymal distribution of tracer showed a significant reduction in *Aqp4* KO mice compared with both CD1 mice ($Aqp4^{+/+}$) and wild-type C57BL/6 (WT) mice (*Figure 4a–b*). No significant difference in global tracer influx was seen between the two control groups. To evaluate if CSF influx was globally decreased or only limited to a particular brain region, we analyzed sections at different anterior-posterior coordinates as analyzed in *Smith et al. (2017)* (*Figure 4c*). Differences in CSF influx between *Aqp4* KO and controls was most notable in coronal sections anterior to bregma (*Figure 4d*), perhaps reflecting that these sections include the cortical segment of the middle cerebral artery, a main site of CSF influx. Regional quantification showed that this difference was the result of decreased tracer penetration in the hypothalamus and areas of the basal forebrain (*Figure 4e–f*). Interestingly, CSF tracer entry along the ventral and lateral cortex was highest in the WT compared to the $Aqp4^{+/+}$ control (*Figure 4f*) suggesting possible strain-dependent differences. We conclude that it is possible to replicate the original findings presented in Iliff et al. in the mouse line reported by Smith et al. when using similar methodology.

## URMC: cerebrospinal fluid entry to brain occurs along the glymphatic pathway and is facilitated by the presence of AQP4 water channels

To evaluate the entry pathways of CSF to the brain we infused Alexa 647-conjugated bovine serum albumin (BSA) and a Texas Red 3 kDa dextran into the cisterna magna of anesthetized WT and *Aqp4* KO mice. In vivo transcranial optical imaging (*Plog et al., 2018*) of the far-red BSA-647 showed more CSF influx in the WT compared to the KO (*Figure 5a and b*). Tracer evidently entered the brain parenchyma through a network of perivascular spaces of the large cerebral arteries on the pial surface. More tracer could be found on the dorsal cortical surface of WT mice compared to *Aqp4* KO mice. After 30 min, brains were extracted and immersion-fixed. The tracer distribution was quantified in coronal sections of both WT and *Aqp4* null mice. The KO mice had less influx for both the 3 kDa dextran and the 66 kDa BSA compared to the WT (~32.7% and ~47.06%, respectively), despite the 22x difference in molecular weight (*Figure 5c–e*). In a replicate experiment where the injection volume and injection rate was half of the previous, A*qp4* deletion also reduced CSF entry through the glymphatic pathway by ~42.4% (*Figure 5f and g*). The relative suppression of CSF tracer influx observed is comparable across previously reported findings, despite use of different tracer molecules and injection paradigms (*Iliff et al., 2012*). The depth of tracer penetration from the cortical surface was quantified using a similar approach to that reported in *Smith et al. (2017)* (*Figure 5f and h*). Tracer penetration into brain was found to be ~2 fold higher in the WT than in the KO mice (*Figure 5i*). To ensure that the mice were in fact knockouts, mice were genotyped using qPCR for both the *Aqp4* locus (*Figure 5j*) and the excision sequence (*Figure 5k*). The results confirmed that all mice were either homozygous for the *Aqp4* locus ($Aqp4^{+/+}$) or were homozygous knockouts ($Aqp4^{-/-}$) possessing no copies of exon 1–3 of the *Aqp4* gene.

## OHSU: reduced glymphatic CSF influx in *Snta1* KO mice with loss of polarized expression of AQP4 in vascular endfeet of astrocytes

To assess the role of perivascular astrocytic localization of AQP4 in mediating CSF flux into the brain parenchyma, we utilized the *Snta1* KO mouse line. These mice lack expression of the adaptor protein

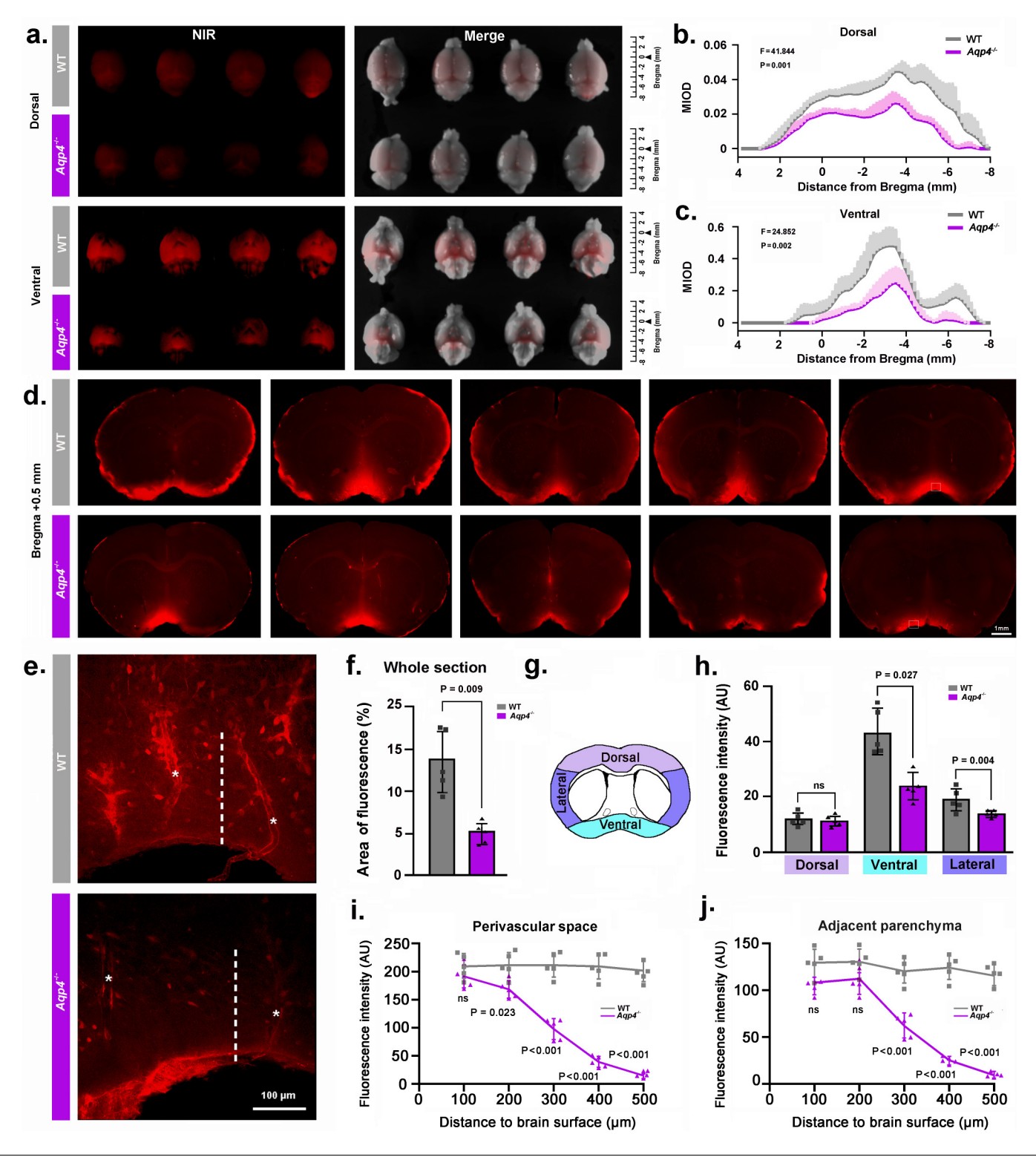

**Figure 2.** NMU: *Aqp4* gene deletion reduced the penetration of intracisternally injected tracer into the brain parenchyma. Texas Red-conjugated dextran (TRd3, 3kD) was injected intracisternally into WT and *Aqp4* KO mice. Thirty minutes after injection, the anesthetized animals were perfusion fixed, and the fluorescence was evaluated ex vivo. (a) Representative near infrared (NIR) fluorescence images of the dorsal and ventral whole-brains of four mice per genotype. (b-c) Quantification of the mean integrated optical density (MIOD) of TRd3 on the dorsal (b) and ventral (c) brain surface of WT (grey) and *Aqp4* KO (purple) mice from 4.0 mm anterior to 8.0 mm posterior to bregma. (d) Representative images of coronal brain sections at +0.5 mm

*Figure 2 continued on next page*

*Figure 2 continued*

from bregma from five pairs of WT and *Aqp4* KO mice showing TRd3 distribution within the brain. (**e**) High magnification micrographs of the hypothalamus (lined area in d) showing the fluorescence intensity of TRd3 within the perivascular space (star) and adjacent brain parenchyma (dotted line) of WT mice and *Aqp4* KO mice, respectively. (**f**) Quantification of the percentage area of whole-slice fluorescence of the both genotypes for 6–8 forebrain sections (+1.7 to −0.7 mm from bregma) of each mouse. (**g**) Diagram showing the subregional analysis of brain sections at the level of 0.5 mm anterior to bregma. (**h**) Quantification of the mean fluorescence intensity (AU, arbitrary units) of TRd3 of the dorsal, ventral and lateral brain regions, respectively. (**i-j**) Quantification of the mean fluorescence intensity of TRd3 along the perivascular space and the interstitium adjacent to the vessels under the ventral surface of the hypothalamus of the both genotypes. Shades and error bars represent standard deviation. Data in *Figure 2b and c* were analyzed by repeated-measures ANOVA, N = 4 per group. Data in *Figure 2f,h,i and j* were analyzed by Student's t-test. p-Values shown are comparisons between WT and *Aqp4⁻/⁻*. ns: not significant (*Figure 2—source data 1*).

DOI: https://doi.org/10.7554/eLife.40070.003

The following source data is available for figure 2:

**Source data 1.** Source data for *Figure 2*.
DOI: https://doi.org/10.7554/eLife.40070.004

α-syntrophin, which links AQP4 to the dystrophin associated complex and is critical for maintenance of the perivascular localization of AQP4 (*Jessen et al., 2015*). Immunolabeling of AQP4 illustrates the loss of AQP4 perivascular localization in the *Snta1* KO mice (*Figure 6a–b*). Higher magnification imaging reveals that, while perivascular localization is lost, widespread AQP4 expression is still detectable by immunofluorescence, consistent with a previous characterization of the mouse line

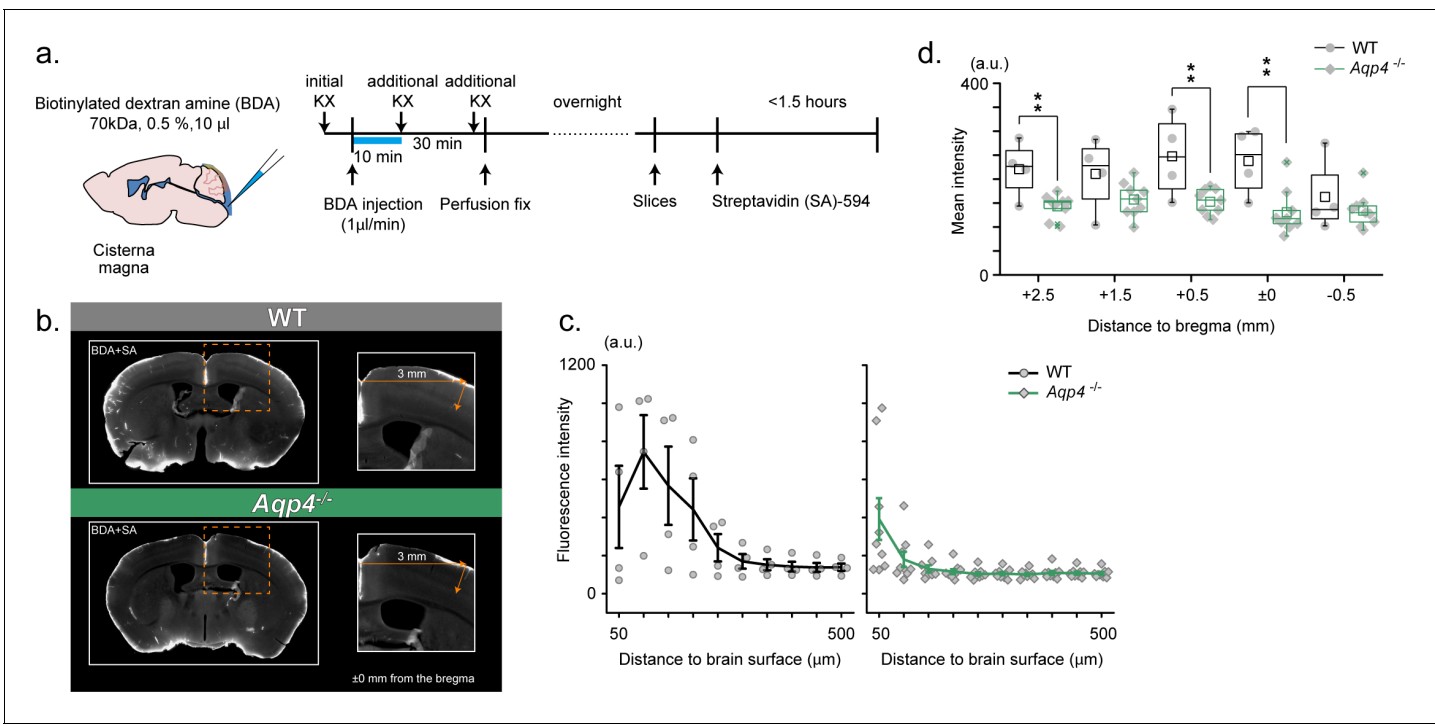

**Figure 3.** RIKEN: *Aqp4⁻/⁻* mice display compromised CSF tracer infiltration under ketamine-xylazine anesthesia. (**a**) Schematic diagram for CM injection of BDA tracer (left) and experiment schedule (right). (**b**) Examples of SA-enhanced BDA distribution 30 min after CM injection. Slices at an anterior-posterior position of bregma are presented for WT (upper) and *Aqp4⁻/⁻* (lower) mice. Depth profile is calculated for the cortical position 3 mm lateral to the midline. (**c**) Mean depth profiles of SA-enhanced BDA signals for WT (black, N = 4) and *Aqp4⁻/⁻* (green, N = 9) mice. (**d**) Mean SA-enhanced BDA signal intensities (3 mm lateral to the midline, depths 0–800 µm) along anterior-posterior positions for WT (black) and *Aqp4⁻/⁻* (green) mice. Shades and error bars represent SEM. *p < 0.05, **p < 0.01, t-test (*Figure 3—source data 1*).

DOI: https://doi.org/10.7554/eLife.40070.005

The following source data is available for figure 3:

**Source data 1.** Source data for *Figure 3*.
DOI: https://doi.org/10.7554/eLife.40070.006

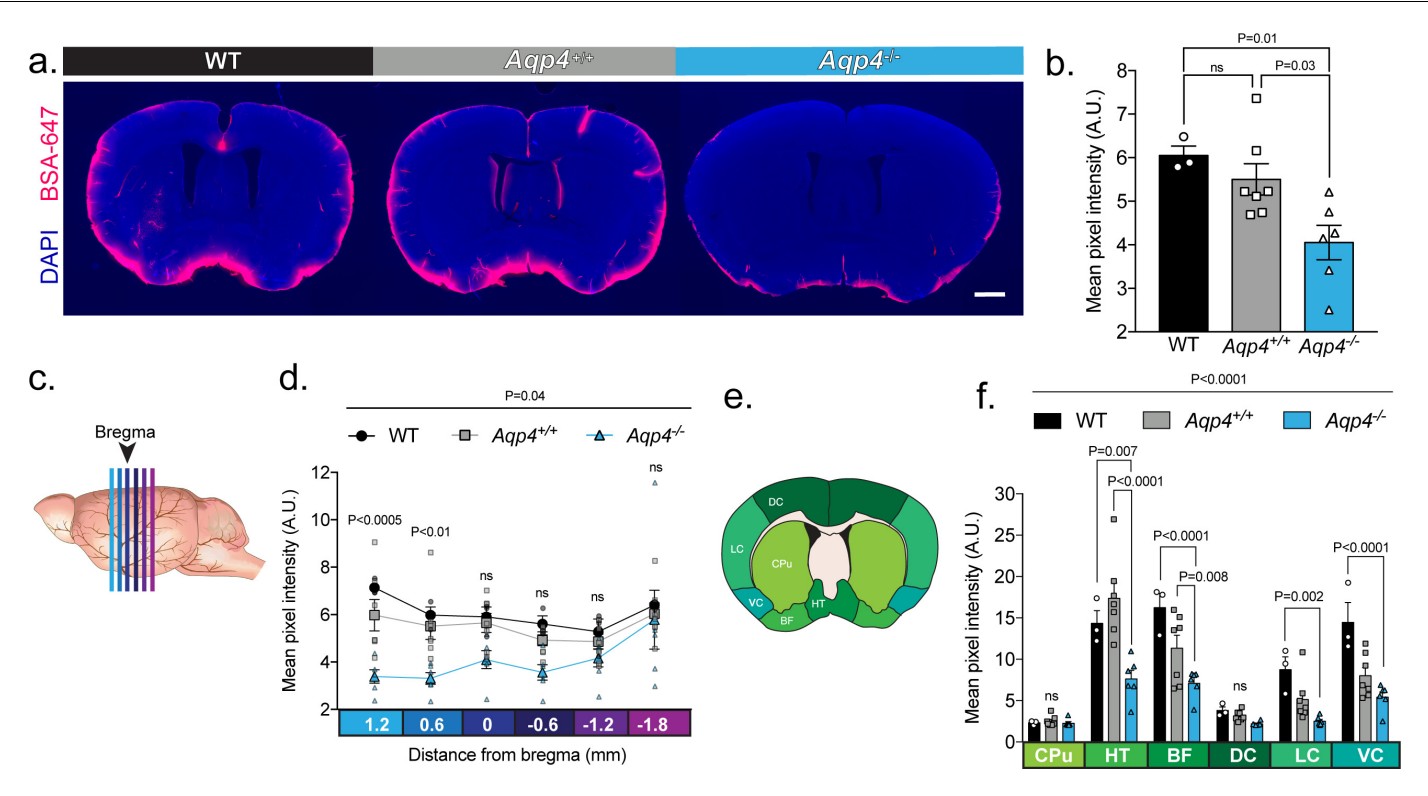

**Figure 4.** UNC: CSF tracer influx is decreased in *Aqp4* KO mice. (**a**) Coronal sections from a C57BL/6 wild-type mouse (WT), CD1 background strain control (*Aqp4*[+/+]), and *Aqp4* KO mice (*Aqp4*[-/-]) showing a fluorescent CSF tracer, BSA-647 and co-labeling with DAPI. Scale bar: 1 mm (**b**) Mean pixel intensity in arbitrary units (A.U.) for six brain sections of each mouse for all three groups. n = 3 (WT), 7 (*Aqp4*[+/+]), 6 (*Aqp4*[-/-]). One-way ANOVA Tukey's multiple comparisons test, Interaction term: p = 0.0110, F = 6.512, ns: not significant. (**c**) Diagram showing the anterior-posterior range of the quantified coronal sections relative to bregma from (**b**). (**d**) Quantification of the slices shown in (**c**) + 1.2 to −1.8 mm from bregma. Repeated measures two-way ANOVA with Tukey's multiple comparisons test, Interaction term: p = 0.038, F = 2.085, p values shown are comparisons of WT and *Aqp4*[+/+] vs. *Aqp4*[-/-]. (**e**) Diagram depicting the ROIs included in the regional analysis of brain slices at +0.6 mm from bregma. CPu: caudoputamen; HT: hypothalamus; BF: basal forebrain; DC: dorsal cortex; LC: lateral cortex; VC: ventral cortex. (**f**) Mean pixel intensity of brain regions shown in (**e**) for coronal sections + 0.6 mm from bregma. Repeated measures two-way ANOVA Tukey's multiple comparisons test, Interaction term: p < 0.0001, F = 8.109. Data is presented as mean ±SEM (*Figure 4—source data 1*).

DOI: https://doi.org/10.7554/eLife.40070.007

The following source data is available for figure 4:

**Source data 1.** Source data for *Figure 4*.

DOI: https://doi.org/10.7554/eLife.40070.008

(*Figure 6a–b* insets (*Neely et al., 2001*)). We next sought to determine if parenchymal CSF influx kinetics are altered when perivascular AQP4 localization is lost. Here, we used dynamic contrast-enhanced magnetic resonance imaging (DCE-MRI) to measure the influx of the contrast agent gadoteridol (Gad) into brain parenchyma after intracisternal injection. We performed serial T1-weighted imaging at 10 min intervals following administration of the contrast agent (*Figure 6c–h*). At 30 min after the start of the injection, elevated levels of Gad were detected in both WT mice and *Snta1* KO mice, particularly along the ventral surface of the brain (*Figure 6e–f*), but by 60 min Gad signal within the parenchyma of *Snta1* KO mice was significantly reduced compared to wild type mice (*Figure 6g,h,n*). Regional assessment revealed decreased signal in *Snta1* KO mice across brain regions, which was most pronounced in cortex and hippocampus compared to subcortical brain regions and within the ventricles (*Figure 6j–m*). Taken together, these data illustrate a role for α-syntrophin in facilitating CSF-ISF exchange and, by extension, support the reported role of AQP4 in this process. Furthermore, these results suggest that the perivascular localization of AQP4 contributes to the kinetics of CSF influx into the brain parenchyma.

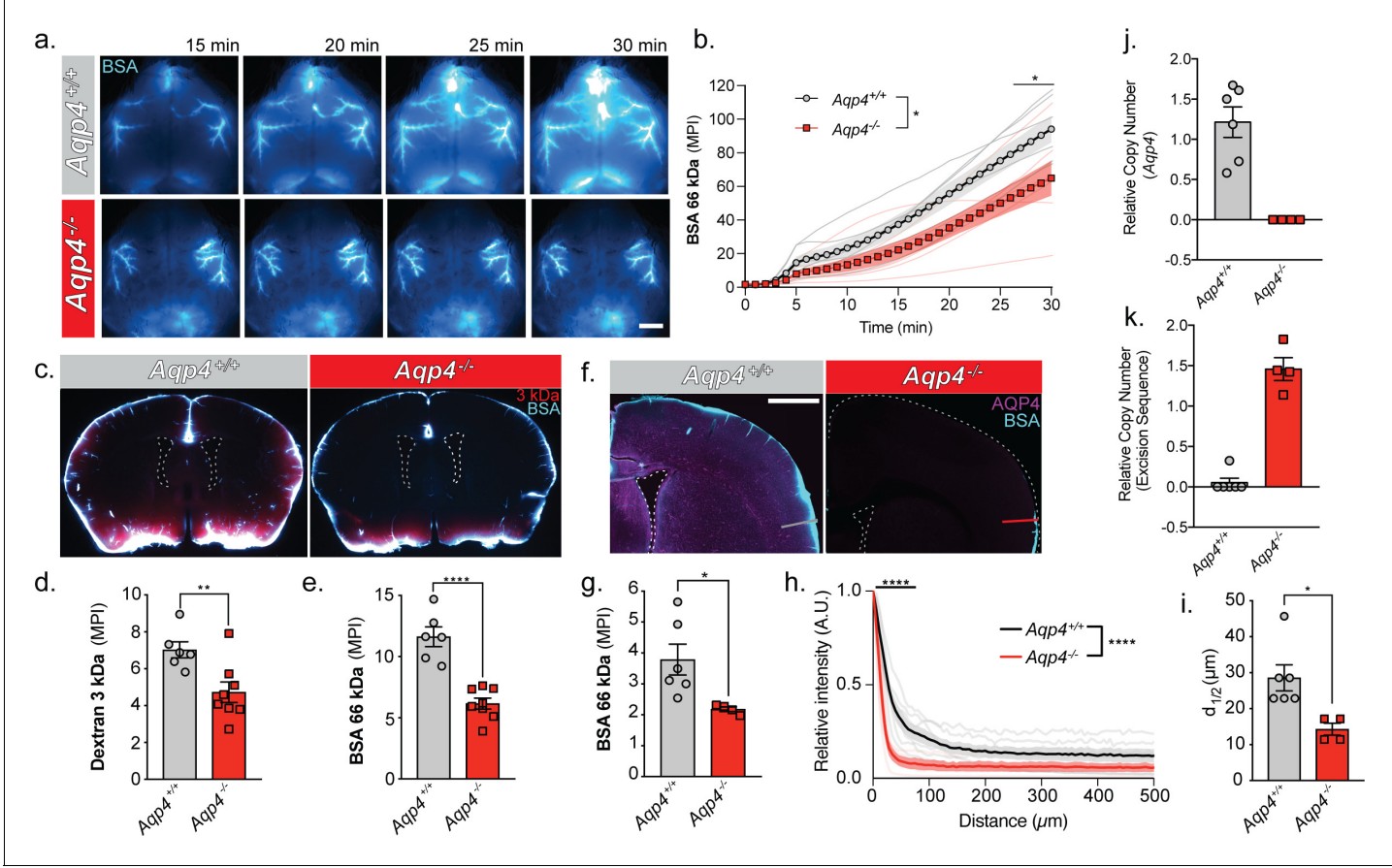

**Figure 5.** URMC: Glymphatic influx of CSF tracer is facilitated by AQP4. (a) Representative images from an in vivo transcranial optical imaging experiment of a WT control mice (*Aqp4*+/+) and *Aqp4* KO (*Aqp4*-/-) mice starting at 15 min after intracisternal delivery of 10 µl (2 µl/min) of a 66 kDa fluorescent tracer, BSA-647. Scale bar: 2 mm. (b) Mean pixel intensity (MPI) of BSA-647 over a 30 min experiment, imaging was started at the beginning of the injection. Two-way repeated measures ANOVA with Sidak's multiple comparisons test, overall model *p = 0.0329, multiple comparisons *p < 0.05; n = 6–7/group. (c) Coronal sections collected 30 min after intracisternal injection of two varying molecular size tracers, 3 kDa Texas Red dextran and BSA-647. Mean pixel intensity for a (d) 3 kDa dextran and (e) BSA from six coronal sections between +1.2 and −1.8 mm from bregma for each animal. Two-tailed unpaired t-test, **p = 0.0097, ****p < 0.0001; n = 6–8/group. (f) Coronal section from an *Aqp4*+/+ mouse stained and imaged for AQP4 (magenta) and BSA-647 tracer (cyan). (g) Mean pixel intensity from a replicate experiment quantified the same as (e). Two-tailed unpaired t-test, *p = 0.0335, n = 4–6/group. (h) Mean tracer penetration depth profiles normalized to the fluorescence at the pial surface of a coronal section at bregma from the set of experiments displayed in (g). The line was placed orthogonal to the cortical surface at the most dorsal position where tracer could be found at the pial surface (f). Tracer depth in the WT mice was measured at the same position as the KO mouse. Two-way repeated measures ANOVA with Sidak's multiple comparisons test, overall model ****p < 0.0001, multiple comparisons ****p < 0.0001, n = 4–6/group. (i) Cortical depth at which the fluorescence at the surface decreases by half for the profiles in (h). Two-tailed unpaired t-test, *p = 0.0161, n = 4–6/group. (j) Relative copy numbers (RCN) of the *Aqp4* gene locus and the excision sequence (k) showing successful deletion of *Aqp4* exon 1–3. RCN was quantified by qPCR for mice used in the experiment in (g). Data expressed as mean ±SEM (*Figure 5—source data 1*).

DOI: https://doi.org/10.7554/eLife.40070.009

The following source data is available for figure 5:

**Source data 1.** Source data for *Figure 5*.

DOI: https://doi.org/10.7554/eLife.40070.010

## Insertion of an injection pipette in striatum is linked to a global suppression of CSF influx

*Smith et al. (2017)* questioned the existence of bulk flow within the brain parenchyma based on imaging tracer dispersion following local injections in cortex or striatum. The tracers were delivered by glass pipettes inserted through a cranial burr hole. The surgical procedure associated with insertion of glass pipettes is obviously traumatic. Since acute and chronic traumatic injury markedly

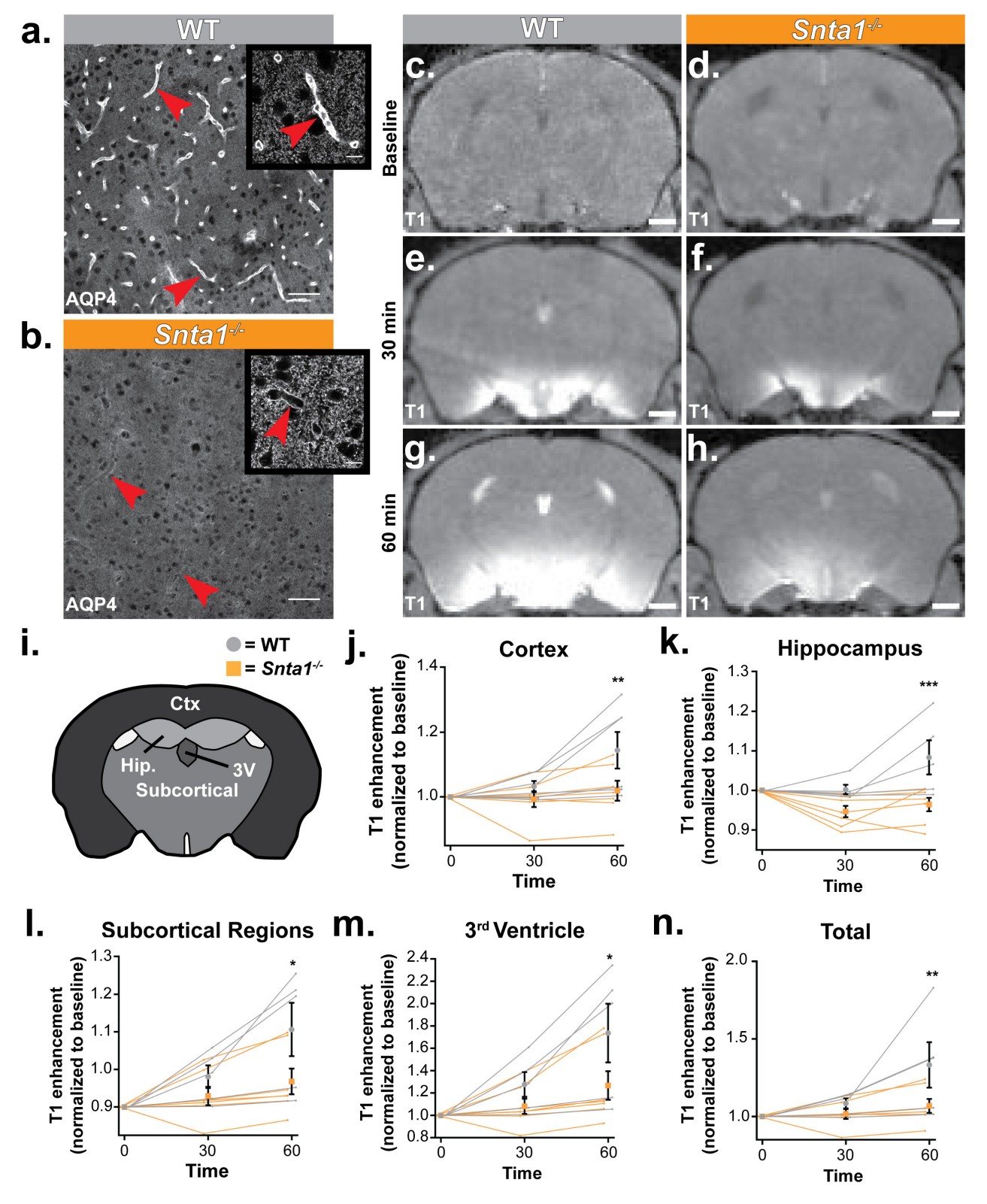

**Figure 6.** Deletion of the adapter protein α-syntrophin impairs AQP4 perivascular localization, and CSF influx into the brain parenchyma. Dynamic contrast enhanced magnetic resonance imaging (DCE-MRI) was acquired on a 11.75 preclinical MRI scanner, and was used to characterize the effect of α-syntrophin deletion on gaditeridol influx into the brain. Representative images of AQP4 perivascular localization in wild-type mice (a), and the loss of perivascular localization of AQP4 seen in the *Snta1*[-/-] mice (b). Scale bar: 50 μm, inset scale bar: 10 μm. (c-h) Coronal slice of $T_1$-weighted images

*Figure 6 continued on next page*

*Figure 6 continued*

acquired by DCE-MRI demonstrate the reduced influx of gaditeridol contrast agent into the parenchyma in *Snta1*[-/-] mice relative to wild-type mice at 30 and 60 min. Scale bar: 1 mm. (**i-n**) Quantification of $T_1$ weighted signal in various brain subregions normalized to baseline at each time point. Traces for each individual animal are presented (lines) along with the summary statistics (mean ±SEM, two-way ANOVA). WT n = 5, ASYNKO n = 7. CTx = cortex (p = 0.0035) Hip = hippocampus (p = 0.0003) Subcortical = subcortical regions (p = 0.0185) 3V = 3[rd] Ventricle (p = 0.0284) Total (p = 0.0085) (***Figure 6— source data 1***).

DOI: https://doi.org/10.7554/eLife.40070.011

The following source data is available for figure 6:

**Source data 1.** Source data for *Figure 6*.
DOI: https://doi.org/10.7554/eLife.40070.012

suppress CSF influx (*Iliff et al., 2014*), we questioned the validity of studying interstitial tracer dispersion using an invasive approach. To this end, we compared CSF tracer (BSA-647, 66 kDa) influx in three groups of mice anesthetized with KX: (i) controls with no surgery beyond CM injection, (ii) insertion of a glass pipette (12 µm diameter tip) and injection of 20 nl (4 nl/min) HiLyte 488-amyloid-$\beta_{1-40}$ in aCSF, or (iii) same as group two but HiLyte 488-amyloid-$\beta_{1-40}$ was not added to the 20 nl aCSF (***Figure 1f***). One hour later, the BSA-647 tracer was injected into cisterna magna and allowed to circulate for 1 hr before the brain was harvested, immersion-fixed, and the distribution of BSA-647 analyzed in coronal brain sections (***Figure 7a–b***). The analysis showed that the two groups exposed to invasive surgery exhibited markedly suppressed CSF tracer influx compared with the control group (***Figure 7c***). The suppression of CSF tracer influx in animals with insertion of the glass pipettes was symmetric with no difference in tracer influx between the ipsilateral and the contralateral hemisphere (p > 0.05, 3–5 slices per animal, n = 7, Wilcoxon rank-sum test). These observations

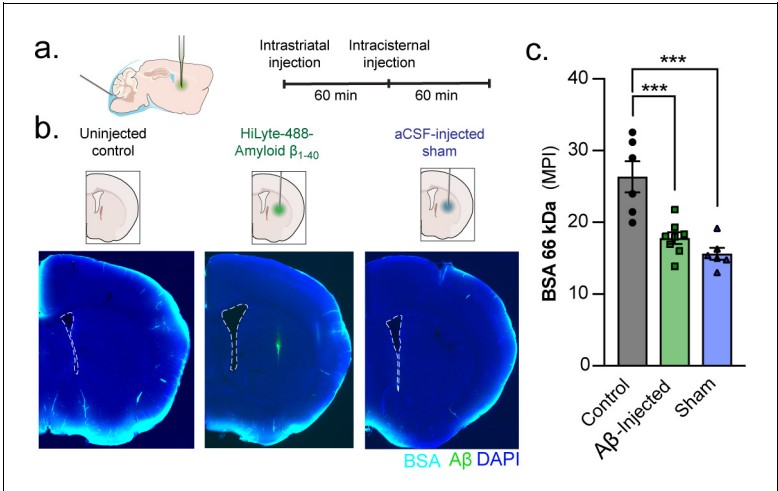

**Figure 7.** A unilateral intrastriatal injection reduces global glymphatic function. (**a**) KX-anesthesized mice received an injection into striatum. After injection, the glass capillary was slowly removed and the skull sealed with silicone elastomer. One hour later, mice received an intracisternal injection of a 66 kDa BSA-647 tracer and brains were removed and drop fixed 60 min later. (**b**) Control mice only received the intracisternal injection but not the intrastriatal injection. Mice that received an intrastriatal injection were injected with either HiLyte 488-amyloid-$\beta_{1-40}$ or an aCSF sham. (**c**) Global glymphatic tracer influx was quantified from a total of six coronal sections between +1.2 and −1.8 mm from bregma for each animal. Ordinary one-way ANOVA posthoc Tukey's multiple comparisons test, Control vs. Aβ-injected: ***p = 0.0006; Control vs. Sham: ***p = 0.0001; n = 6–8/group (***Figure 7—source data 1***).
DOI: https://doi.org/10.7554/eLife.40070.013

The following source data is available for figure 7:

**Source data 1.** Source data for *Figure 7*.
DOI: https://doi.org/10.7554/eLife.40070.014

show that invasive procedures acutely suppresses brain-wide CSF influx and that tracer injection should not be used in the study of the glymphatic system.

## Meta-analysis

A potential cause of conflicting results could be due to varying methodology used in each study. We used meta-analysis to evaluate this heterogeneity and better understand the relationship between AQP4 and CSF-ISF transport using all of the available literature that evaluated tracer injections in global *Aqp4* KO and wild-type mice. Studies were largely divided into two types: 1) intracisternal and 2) intracerebral tracer injections (*Figure 8a–b*), a separate meta-analysis was conducted for each of these two main experimental approaches.

### CSF influx

The search for studies that delivered tracers into the cisterna magna to evaluate CSF entry yielded a total of 13 eligible independent datasets from six different laboratories (*Supplementary file 1*). Experiments were done in a total of 144 mice (92.4%) and rats (7.6%) from five different *Aqp4* KO lines (UIO: 50.7%; UCSF mouse: 25.7%; RIKEN: 9.03%, UCSF rat: 7.64%, and NMU: 6.94%) with a wide age range (6–24 weeks), the age of the rat line was not reported. Results were obtained using three different anesthetic protocols (ketamine/xylazine: 68.8%; tribromoethanol: 24.3%; chloral hydrate: 6.94%) and it was not possible to determine sex due to variable reporting. Experiments consisted of the delivery of fluorescently- (86.1%) and radioisotope-labeled (13.9%) tracers that ranged in size (4.5–70 kDa) and concentration. Tracer delivery varied in total volume (5–10 μl) and the injection rate (1–2 μl/min) while experiment duration ranged from 15 to 30 min. Standardized mean differences (SMD) were used in the meta-analysis to account for different outcome measures (e.g. fluorescence intensity, thresholded area, % injected radiation), and demonstrated a significant overall decrease in tracer entry in *Aqp4* KO mice compared to controls (SMD with 95% confidence interval: −1.70 [−2.49; −0.90]; p<0.0001) with a high degree of interstudy heterogeneity ($I^2$ = 73.1%; p<0.0001; *Figure 8a*). Leave-one-out and meta-regression sensitivity analyses were used to explore the sources of heterogeneity. As expected based on the direction and magnitude of the effect estimates reported in the studies, the high degree of heterogeneity was primarily caused by three datasets extracted from Smith et al. We could not confidently determine from the text of these studies whether these experimental figures were based on separate, independent experiments, or whether they were data collected in the same set of mice. Due to this uncertainty, we conducted leave-one-out analyses that would recapitulate all possible scenarios. The analysis showed that when these data sets were excluded, there was a reduction in heterogeneity ($I^2$ = 65.7%, 70.3%, and 71.4%; p = 0.0007, p = 0.0001 and p < 0.0001). When all three data sets from this publication were excluded, the heterogeneity becomes non-significant ($I^2$ = 0.0%, p = 0.4622). Due to the relevance of this study, we included all data sets in the final results reported (*Supplementary file 1*). Using meta-regression to further explore sources of heterogeneity, we tested whether the: KO line, age, anesthesia type, tracer properties, injection paradigm, experiment duration, and detection method were significant covariates (*Supplementary file 1*). The KO line used (p = 0.0001, Test of Moderators), the age of the mice (p = 0.0006), the anesthetic protocol (p < 0.0001), and the injection rate (p < 0.0001) explained a significant proportion of the observed variance between the datasets. The variables that explained the largest proportion of heterogeneity were anesthesia type and injection rate. The three data sets that reported null results used tribromoethanol (Avertin), an anesthetic that was not used in any other studies (all of which found a significant association). The same three data sets were also the only ones to report variable injection rates (10 μl delivered over 5–7 min) in comparison with the remaining studies that used exact injection rates. Thus, it is possible that the choice of anesthesia type and injection rate could explain the heterogeneity. Animal age was also a significant source of heterogeneity, accounting for 85.28% of the variance between studies. This may be in part explained by the fact that two out of the three data sets that demonstrated no association were completed in mice that were older than the mice used in the majority of other studies and reported the largest age range (12–24 weeks). Although it is not possible to determine the exact role that these covariates played in the observed results, both anesthesia and aging have been shown to modify the degree of intracisternal tracer entry, providing another potential explanation for the observed heterogeneity (*Benveniste et al., 2017*; *Kress et al., 2014*). Similarly, the

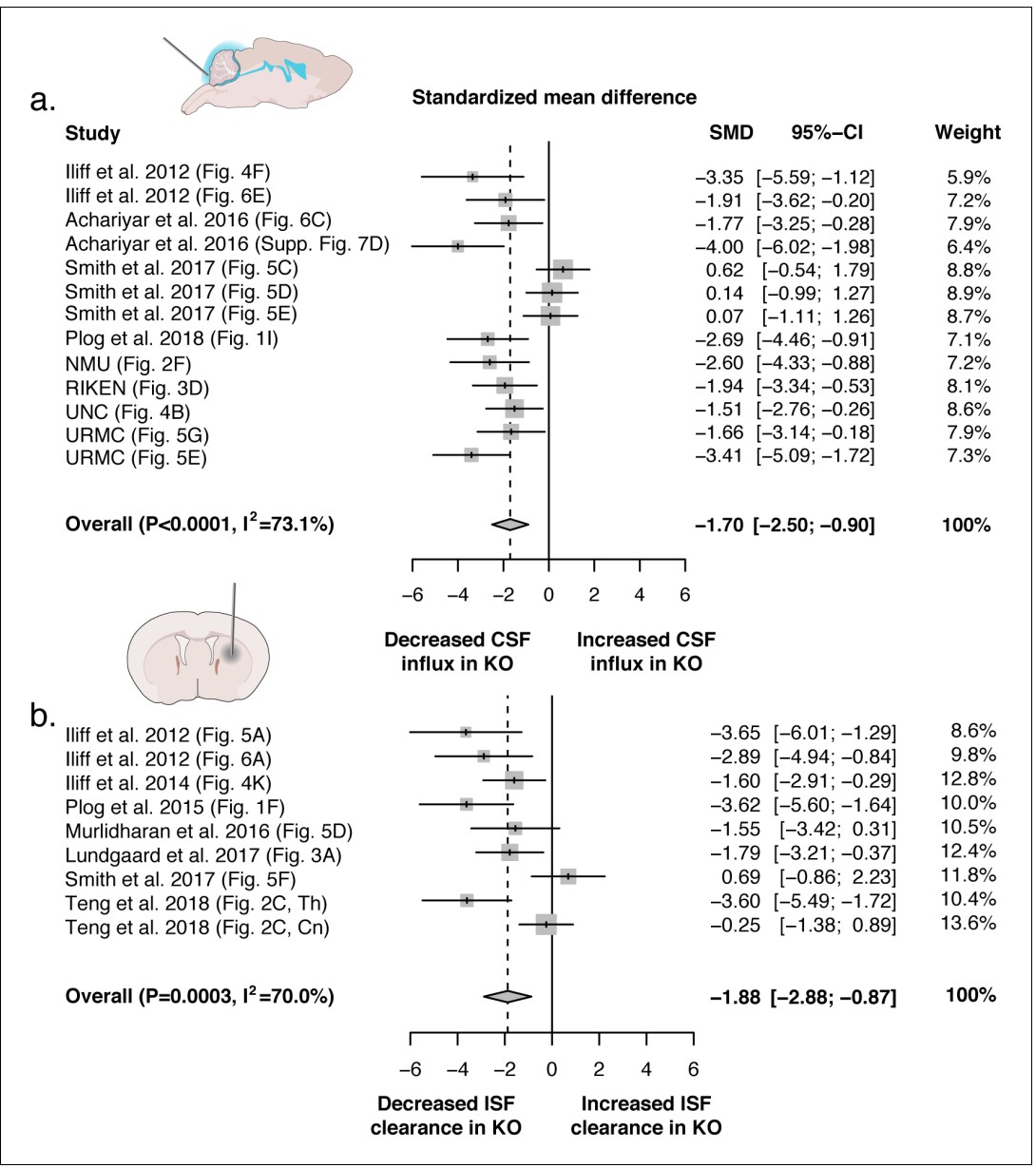

**Figure 8.** Evidence evaluating the role of AQP4 in CSF influx and ISF efflux. (a) Meta-analysis from experiments that delivered either fluorescence- or radio-labeled tracers into the cisterna magna of both *Aqp4* KO and wild-type rodents. (b) Meta-analysis from studies that delivered intracerebral tracers to evaluate clearance or transport of tracers out of the brain. p-Value is from the overall random effects model. Data in forest plots presented as standardized mean difference (SMD) with a 95% confidence interval (CI). Th: thalamus; Cn: caudate nucleus.
DOI: https://doi.org/10.7554/eLife.40070.015

heterogeneity stemming from the KO line may also be explained by the fact that the majority of UCSF KO mice (72.9%) were in the null studies. Interestingly, the fourth study (UNC, this report) that utilized the UCSF KO mouse line, but an alternate anesthetic (ketamine/xylazine) observed a significant effect. Unfortunately, due to low study number we were unable to include two covariates (e.g. anesthesia type and KO line) in the meta-regression to test this interaction. Additional covariates including tracer properties, the injection paradigm, and the detection method were not significant sources of heterogeneity suggesting that multiple experimental methodologies should be able to observe an effect between KO and control.

## ISF efflux

There were nine eligible studies produced by four independent research groups that injected tracers or contrast agents into the brain of *Aqp4* KO or wild-type rodents in order to evaluate brain clearance or ISF efflux (*Supplementary file 2*). These experiments were conducted in a total of 87 rodents from both mouse (72.4%) and rat (27.6%) KO lines. There were two murine lines (UIO: 57.5%; UCSF: 14.9%) and one rat line (Beijing: 27.6%) generated by three separate groups. Age of the mouse lines ranged broadly (0–24 weeks), while the age of the rat line could not be confirmed but a weight range of 250–300 g was reported. Four anesthetic protocols were used for these experiments (ketamine/xylazine: 57.5%; hypothermia: 6.9%; tribromoethanol: 8.0%; sodium pentobarbital: 27.6%). Fluorescence-conjugated tracers (14.9%), radio-labeled (57.5%) tracers, and gadolinium-based contrast agents (27.6%) that ranged in size (0.18–45 kDa), concentration, and where they were delivered within the brain (striatum: 40.2%; cortex: 39.1%; thalamus: 13.8%; lateral ventricle: 6.9%). Tracer delivery varied in total volume (0.02–3 µl) and injection rate (0.017–3 µl/min), differing based on the injection site or whether the experiment was done in mice or rats. Experiment durations ranged between 45 and 240 min and one of the studies (Smith et al.) did not report this. Results from the overall meta-analysis showed a significant decrease in the transport of the injected substance in *Aqp4* KO mice relative to controls (−1.88 [-2.88; −0.87]; p = 0.0003) with high heterogeneity ($I^2$ = 70.0%, p = 0.0008; *Figure 8b*). Similar to the previous model, leave-one-out analysis of the two null datasets reduced the heterogeneity ($I^2$ = 60.1%; p = 0.0143 and $I^2$ = 65.3%; p = 0.0053, respectively) and removing both datasets made the heterogeneity non-significant ($I^2$ = 17.2%; p = 0.299). The same covariates used in the first model were used for meta-regression analysis, in addition to an injection site variable. Meta-regression was unable to identify significant sources of heterogeneity; most likely because of two datasets from Teng et al. that found a difference in thalamus, but not in striatum, in the same rat line using identical methodology (*Teng et al., 2018*). Despite the potential biological significance of this, the study was the only one to use rats, MRI as the detection method, and had an experiment duration that was 4-fold longer than the other studies. Therefore, we decided to run a reduced model on all available mouse experiments using fluorescence or radioactivity detection methods. The meta-analysis was still highly significant and showed decreased clearance in the *Aqp4* KO mouse lines (−1.92 [-3.02; −0.82]; p = 0.0006) with a lower degree of heterogeneity ($I^2$ = 64.1%, p = 0.0103). The KO line used (p = 0.0205), age (p = 0.0075), the anesthetic protocol (p = 0.0075), the injection rate (p = 0.0056), the experiment duration (p = 0.0075), and the detection method (p = 0.0205) explained a significant proportion of the observed variance between the datasets. Similar to previous results, tribromoethanol again seemed to function as a proxy for the null study from Smith et al. since none of the others used this anesthetic. The mouse KO line from UCSF explained a significant proportion of heterogeneity compared to the other lines, likely because the null dataset accounted for half of the studies that used this line. Age of the experimental animals was also a significant source of heterogeneity and can be explained by the wide age range (0–24 weeks) found between the studies but the lack of specific age reporting could also potentially explain this. Injection rate accounted for the largest proportion of heterogeneity in this analysis. The injection rate used by Smith et al. (5 × 100 ms pulses at 1 Hz and 10 psi), was highly significant in the meta-regression model compared to the reference injection rate (0.017 µl/min, p = 0.0007), whereas all other injection paradigms were not significantly different. The Smith paradigm was therefore the driver of the explanatory value of the injection rate covariate in the meta-regression model. Similar to anesthesia type, this variable also functioned as a proxy for the null study since all other datasets reported an exact volumetric injection rate. The reported injection protocol was used for experiments performed in both mice and rats despite different injection volumes (~20 nl vs. 30,000 nl) suggesting that the rate varied significantly between experiments. The time the tracer was allowed to circulate after the injection (experiment duration) was also a highly significant source of variation between studies. The experiment duration that accounted for all the heterogeneity was the unreported time from the null study. Only one separate study evaluated a time point that did not report a significant difference between KO and control (30 min; *Iliff et al., 2012*). However, this study did report a difference at 60 min after injection, so it is possible that if Smith et al. used shorter experimental durations this could account for the observed heterogeneity (*Supplementary file 2*) (*Iliff et al., 2012*). The detection method (i.e. fluorescence and radioactivity) used was also found to be a significant source of heterogeneity. The largest contributor to

heterogeneity were fluorescent detection methods, mostly likely due to the fact that the study that obtained a null result used this approach despite most studies using radiolabeled tracers. The remaining covariates: tracer properties, the injection volume, and the injection site were not significant sources of heterogeneity in this analysis.

## Discussion

This study evaluated the role of the astrocytic water channel AQP4 in the dispersion of fluorescent tracers or contrast agents by five independent groups. Four of the groups (NMU, RIKEN, UNC, URMC) injected CSF tracers in cisterna magna and compared the influx of the tracer in coronal sections prepared from *Aqp4* KO and WT mice, whereas the OHSU group injected contrast agent in cisterna magna and then compared CSF transport by DCE-MRI in *Snta1* KO and WT mice. Compilation of data from the six independently-conducted experiments documented that deletion of *Aqp4* (*Aqp4* KO mice) or mis-localization of AQP4 (through *Snta1* gene deletion) suppressed influx of CSF tracers or contrast agents. In fact, a meta analysis that included all reports published to date came to the same conclusion (*Figure 8*). Thus, our observations provide strong and concordant support for the glymphatic model in which AQP4 supports the perivascular influx of CSF and efflux of ISF (*Iliff et al., 2012*). Supporting evidence of this has also been shown in humans; two independent single nucleotide polymorphism (SNP) variant association studies identified multiple SNPs in the *Aqp4* gene that were associated with cognitive decline in Alzheimer's disease (AD) and increased Aβ burden in humans with reduced sleep quality (*Burfeind, 2017*; *AIBL Research Group et al., 2018*). A separate study by the OHSU group also found that reduced perivascular polarization of AQP4 was a significant predictor of AD status in postmortem human brains (*Zeppenfeld et al., 2017*).

Independent replication studies of biological phenomena are essential, both to confirm novel experimental findings, as well as to extend and refine initial observations and interpretations. The original report from the URMC group on the glymphatic system showed that deletion of astrocytic water channels suppressed both the influx of CSF tracers injected into the cisterna magna and the efflux of tracers infused directly into striatum (*Iliff et al., 2012*). Follow-up of that finding by the NMU group documented that Aβ accumulation and cerebral amyloid angiopathy were significantly increased in a double-hit murine model with *Aqp4* deletion and Aβ over-expression (*Xu et al., 2015*). The URMC/OHSU groups subsequently reported that deletion of *Aqp4* aggravated neurofibrillary pathology following TBI (*Iliff et al., 2014*), whereas the UNC group showed that deletion of *Aqp4* significantly reduced clearance of AAV (*Murlidharan et al., 2016*). Thus, these reports conceptually address the same idea, although the replications entailed different approaches and methodologies. The dissenting report from Dr. Verkman's laboratory belongs to the rarer category of replication studies, which (as their title states) seeks to test directly the glymphatic hypothesis and the AQP4-dependence of solute transport in brain. Since their study was intended to gauge the veracity of the original findings, it should be expected that it follow the experimental design and data analysis of the original study (*Picho et al., 2016*), and their methodology should be presented in sufficient detail to allow the reader to assess the fidelity of their replication (*Brandt et al., 2014*). However, *Smith et al. (2017)* failed to meet these requirements in several key respects. First, their methodologies deviate from the original report in critical ways. In particular, Iliff et al. used ketamine/xylazine (KX) anesthesia rather than tribromoethanol (Avertin) (*Iliff et al., 2012*). Xylazine is an $\alpha_2$ adrenergic agonist that blocks release of norepinephrine from the locus coeruleus projections across the neuraxis. This is an important point, because norepinephrine has been identified as the key humoral regulator of glymphatic solute transport (*Xie et al., 2013*), and norepinephrine attenuates glymphatic function. However, with the exception of the photo-bleaching experiments carried out only in WT mice, *Smith et al. (2017)* anesthetized their mice with Avertin, an injectable anesthetic with an unknown mechanism of action. Avertin is not approved for use in several European countries and is restricted to terminal procedures in the US (*Arras et al., 2001*; *Meyer and Fish, 2005*). The importance of the anesthetic agent in the present context was first highlighted by *Groothuis et al. (2007)*, who reported that parenchymal solute efflux was 100-fold slower in rats anesthetized with pentobarbital than those that received ketamine/xylazine. Anesthesia-dependent effects were also noted by Benveniste et al., that reported a marked difference in influx of contrast agents in isoflurane-anesthetized animals versus low-dose isoflurane plus dexmedetomidine (an $\alpha_2$ agonist) (*Benveniste et al., 2017*). The importance of anesthesia is supported by the experiments

reported here: The UNC group used ketamine/xylazine anesthesia and observed AQP4-dependent glymphatic transport (*Figure 4*), despite using the same *Aqp4* KO mouse line as in the *Smith et al. (2017)*. In addition, the methodology of Smith et al. was not presented in sufficient detail to assess the fidelity of the replication. Key methodological details, such as the injection approach and rate, anesthesia, or the age of the animals were not provided for data collected in rats. Neither did the investigators assess whether *Aqp4* was deleted in CNS albeit this was the first report utilizing *Aqp4* KO rats. In fact, no such data have been published so far to our knowledge (*Smith et al., 2017*). For these reasons, the present study is not a replication study of *Smith et al. (2017)*. Instead, we used the methodology reported in *Iliff et al. (2012)* and further optimized it in subsequent studies (*Kress et al., 2014*; *Iliff et al., 2013a*).

It is also important to note that the recovery period that follows general anesthesia differs from the awake state. A recent study reported that increasing anesthesia concentration decreased glymphatic function (*Gakuba et al., 2018*), as opposed to the original finding that the glymphatic system is maximally active during natural sleep or anesthesia (*Xie et al., 2013*). However, the recent study injected CSF tracers under inhalation anesthesia followed by a short recovery period (*Gakuba et al., 2018*). This is not a valid approach, because awakening from general anesthesia is associated with cognitive disturbances (*Su et al., 2016*; *Paredes et al., 2016*; *Numan et al., 2017*) and inhaled anesthetics in particular are linked to a high rate of delirium upon emergence from anesthesia (*Brioni et al., 2017*). Close evaluation of the imaging in Gakuba et al. shows a failed intracisternal injection, as enhancement of contrast agent is evident in the paraspinal muscles and extracranial spaces (*Gakuba et al., 2018*). In our experience, experimenter error during cisterna magna injections is a significant source of variability (*Xavier et al., 2018*). Thus, careful consideration of the anesthesic regimen and long periods of post-anesthesia recoveries are essential when studying the glymphatic system.

The initial study articulating the glymphatic model confirmed seminal work by Cserr and colleagues that interstitial tracers are cleared from brain tissue at rates that are independent of molecular weight (*Iliff et al., 2012*; *Cserr et al., 1981*). While these early studies support the role of bulk flow in the clearance of interstitial solutes from the brain interstitium, they lacked the spatial resolution necessary to resolve whether bulk flow was occurring throughout the entire brain interstitium or was restricted to a subset of compartments permissive to flow. A large body of indirect experimental evidence supports the occurrence of bulk flow in the periarterial space in the brain (see reviews by *Hladky and Barrand, 2014* and *Nicholson and Hrabětová, 2017*). *Smith et al. (2017)* themselves provide such evidence, stating that 'occasionally, when the injected dextran entered the perivascular space of vessels near the injection site, both large and small dextrans traveled substantial distances away from the injection site' (*Smith et al., 2017*). There is also indirect evidence that this flow is peristaltic, propelled by pulsations of the arterial walls (e.g. *Hadaczek et al. (2006)*). This mechanism has recently been confirmed by direct observations of pulsatile flow in periarterial spaces (*Bedussi et al., 2018*; *Mestre et al., 2018*). Using two-photon microscopy and particle-tracking velocimetry, Mestre et al. gathered hundreds of thousands of velocity measurements at sampling rates up to 30 Hz over half hour experiments, mapping the perivascular flow in space and time in great detail: their results show that the flow pulses in synchrony with the heartbeat, is driven by pulsations of the arterial wall, and produces a net (average) flow in the same direction as the blood flow. This flow is in accordance with several earlier fluid-dynamic models and experimental studies (*Bilston et al., 2003*; *Schley et al., 2006*; *Wang and Olbricht, 2011*; *Rennels et al., 1985*; *Iliff et al., 2013b*), and with in vivo dynamic imaging studies, including MRI experiments in human subjects, demonstrating the rapid distribution of CSF tracers into and through the brain interstitium along peri-arterial routes (*Iliff et al., 2013a*; *Eide and Ringstad, 2015*; *Ringstad et al., 2017*). In addition to perivascular spaces, white matter tracts also serve as permissive routes for interstitial solute and fluid movement (*Ohata and Marmarou, 1992*; *Rosenberg and Kyner, 1980*).

As a consequence of the now well-established bulk flow along periarterial spaces, there must be an outlet for this directional flow: conservation of mass - as expressed by the continuity equation - demands that this volumetric flow must continue through other parts of the glymphatic system – that is, in the neuropil and the perivenous spaces. Although the full volumetric net flow must pass through the neuropil, flow speeds will be lower there because of the much larger total cross-sectional area of the available extracellular flow channels, and due to the availability of the gap junction-coupled astroglial syncytium as an intracellular route for water and solute

movement (*Asgari et al., 2016*). In fact, a meta-analysis that include the more than 200 *Aqp4* KO mice and rats studied so far added further support to the finding that glymphatic influx and efflux are both highly dependent upon AQP4 expression (*Figure 8* , *Supplementary file 1* and *2*). The meta-analysis identified the anesthesia protocol, wide age range of the experimental animals and the injection paradigm are potential causes for the conflicting data reported by Smith et al. However, it is important to note that glymphatic transport is affected by multiple pathways other than AQP4. For example, the sleep-wake cycle (*Xie et al., 2013*), brain injury in the setting of trauma or ischemia (*Iliff et al., 2014*; *Wang et al., 2017*), exercise (*He et al., 2017*; *von Holstein-Rathlou et al., 2018*; *Yin et al., 2018*), amyloid-β accumulation and acute amyloid-β toxicity (*Xu et al., 2015*; *Peng et al., 2016*), omega-3 fatty acids (*Ren et al., 2017*), plasma osmolarity (*Plog et al., 2018*), and aging (*Kress et al., 2014*) are important regulators of glymphatic transport. How AQP4 on a cellular level facilitates CSF-ISF exchange remains to be firmly established. The primary role of AQP channels is to lower the driving forces for fluid transport across the plasma membranes (*Verkman et al., 2017*). AQP4 is highly enriched in astrocytic endfeet facing the perivascular space and thus strategically positioned to facilitate CSF exchange with interstitial fluid. Accordingly, loss of the polarized expression of AQP4 in reactive astrocytes lead to a reduction in parenchymal CSF tracer influx (*Iliff et al., 2014*; *Kress et al., 2014*). Moreover, deletion of AQP4 potentiated the increases in intracranial pressure in a murine model of hydrocephalus consistent with the notion that AQP4 supports and facilitates intraparenchymal fluid flow (*Bloch et al., 2006*).

The claim by *Smith and Verkman (2018)* that solute transport in the parenchyma is primarily driven by diffusion rather than convection (bulk flow) is based on their finding of a dependence of this transport on the size of the solute particles. They claim that diffusive transport depends strongly on particle size, whereas convective transport does not, so long as the ratio of the particle size to the pore or channel size is less than about 0.5. In support of this claim, they cite a single reference (*Dechadilok and Deen, 2009*), which does not actually support their claim, but instead shows a significant dependence of flow resistance on particle size for much smaller particles. The influence of solutes or suspended particles on flow through narrow channels is a complicated problem in hydrodynamics, entailing many factors, including solute concentration, electrical charge, shape of the particle, channel geometry, tortuosity, and possible obstructions (*Nicholson and Hrabětová, 2017*). Hence, it would require a very sophisticated theoretical model to evaluate the role of bulk flow in the transport observed experimentally by *Smith et al. (2017)*. Their findings are in general agreement with recent modeling studies suggesting that at the smallest microscopic scales, encompassing small numbers of vessels and the interposed neuropil, diffusion is sufficient to account for observed solute movement (*Asgari et al., 2016*; *Jin et al., 2016*; *Holter et al., 2017*). However, it is important to note that these findings do not exclude a contribution of bulk flow within the entire interstitium, and do not determine whether the experimental approaches used to detect flow on the microscopic scale are sufficiently sensitive to detect bulk flow occurring at very low flow rates. Also, the majority of the studies contradicting the glymphatic model are based on theoretical modeling and not experimental evidence (*Asgari et al., 2016*; *Jin et al., 2016*; *Faghih and Sharp, 2018*). We suspected that the surgical procedures required for intraparenchymal tracer injections utilized in *Smith et al. (2017)* might suppress influx of CSF tracers. To formally test this potential caveat, CSF tracer influx was analyzed in mice in which a glass pipette was inserted in striatum 1 hr earlier (*Figure 7*). The analysis showed that insertion of a glass pipette suppressed CSF tracer influx significantly and independently of intrastriatal injection of aCSF with or without HiLyte 488-amyloid-$\beta_{1-40}$ compared to non-surgical controls. We conclude that dispersion of tracers in CNS should not be studied using invasive procedures. Thus, as articulated by Holter et al. in their recent modeling study (*Holter et al., 2017*), there is a need to develop more appropriate technical approaches for evaluating slow interstitial bulk flow over long distances (*Ratner et al., 2017*), such as DCE-MRI in nonhuman primates or human subjects. Until then, the glymphatic model of CNS fluid and solute flow should properly include the rapid exchange of CSF with the brain ISF along perivascular spaces, with CSF principally entering along peri-arterial spaces and ISF draining towards the ventricular and CSF compartments along peri-venous pathways and along cranial and spinal nerves, supported by AQP4-dependent astroglial water transport. On this last element, the present findings from five independent groups do not concur with the negative findings reported by *Smith et al. (2017)*. Instead, our compiled data provides strong and consistent support for our claim that solute transport in the rodent brain is facilitated by the polarized expression of AQP4 in astrocytic endfeet.

# Materials and methods

**Key resources table**

| Reagent type (species) or resource | Designation | Source or reference | Identifiers | Additional information |
|---|---|---|---|---|
| Gene (*Mus musculus*) | Aqp4 | | MGI:107387 | |
| Gene (*Mus musculus*) | Snta1 | | NCBI_gene: 20648, MGI:101772 | |
| Strain, strain background (*Mus musculus*) | *Aqp4* null/eGFP knock-in mice | RIKEN BRC | RBRC06500; SJL.Cg-Aqp4<tm1.1(GFP) Mysi>/MysiRbrc; CDB0758K-1 | |
| Strain, strain background (*Mus musculus*) | *Aqp4* knockout mice | PNAS 108, 846–851 (2011). PMID: 21187412 | OIU | |
| Strain, strain background (*Mus musculus*) | C57BL/6N | Beijing Vital River Laboratory Animal Technology Co., Ltd. | 213 | |
| Strain, strain background (*Mus musculus*, C57Bl/6J) | male wild type (WT) | Jackson Laboratories, 000664 | RRID:IMSR_JAX:000664 | |
| Strain, strain background (*Mus musculus*, C57BL/6NCrl) | male wild type (WT) | Charles River, 027 | RRID:IMSR_CRL:27 | |
| Strain, strain background (*Mus musculus*, C57Bl/6J) | male Snta1$^{-/-}$ | Jackson Laboratories, 012940, PMID:10995443 | RRID:MGI:2181419 | |
| Cell line (*Mus musculus*) | E14K ES | J Neurosci Res. 2005 Nov 15;82 (4): 458–64. PMID: 16237719 | 129P2/OlaHsd | |
| Transfected construct (*Mus musculus*) | AQP4 replacement targeting vector | J Neurosci Res. 2005 Nov 15;82(4): 458–64. PMID: 16237719 | | Constructed using the positive–negative selection cassettes derived from the vectors pPolII long neo bpA and pXhoMC1TK, containing the neoR and HSVtk genes, respectively. |
| Biological sample (*Mus musculus*) | CD1 blastocysts | Beijing Vital River Laboratory Animal Technology Co., Ltd. | 201 | |
| Antibody | anti-Aqp4 (rabbit polyclonal) | Sigma | A5971, RRID:AB_258270 | (1:1000) |

*Continued on next page*

*Continued*

| Reagent type (species) or resource | Designation | Source or reference | Identifiers | Additional information |
|---|---|---|---|---|
| Antibody | biotinylated-conjugated goat anti-rabbit IgG (polyclonal) | Vector Laboratories | BA-1000, RRID:AB_2313606 | (1:200) |
| Antibody | AQP4 (rabbit primary Ab) (polyclonal) | Millipore | AB3594, RRID:AB_91530 | (1:400–500) |
| Antibody | Alexa Fluor 594 (donkey anti-rabbit secondary Ab) (polyclonal) | ThermoFisher | A21207, RRID:AB_141637 | (1:500) |
| Antibody | Cy3-conjugated donkey anti-rabbit (polyclonal) | Jackson Immuno Research | 711-165-152, RRID:AB_2307443 | (1:500) |
| Sequence-based reagent | oligonucleotide primers of Aqp4 and excision sequence | Transnetyx | Probe ID: Aqp4-2 WT and Aqp4-2 KO | |
| Sequence-based reagent | oligonucleotide primers of Aqp4 and Neo | J Neurosci Res. 2005 Nov 15;82 (4) :458–64. PMID: 16237719 | | AQP4 forward primer, ACC ATA AAC TGG GGT GGC TCA G; WT AQP4 reverse primer, TAG AGG ATG CCG GCT CCA ATG A; and Neo, CAC CGC TGA ATA TGC ATA AGG CA. |
| Peptide, recombinant protein | Beta - Amyloid (1 - 40), HiLyte Fluor 488 - labeled, Human | Anaspec | AS-60491–01 | 0.5% in aCSF |
| Commercial assay or kit | Kwik-Cast | World Precision Instruments | KWIK-CAST | |
| Commercial assay or kit | Elite ABC Kit | Vector Laboratories | PK-7200 | |
| Chemical compound, drug | BDA | Molecular Probes/Thermo Fisher | D1957; BDA-70,000 | |
| Chemical compound, drug | Alexa-594-conjugated streptavidin (SA) | Molecular Probes/Thermo Fisher | S11227 | (1:1000) |
| Chemical compound, drug | Alexa Fluor 647 conjugated bovine serum albumin | Molecular Probes/Thermo Fisher | A34785 | 0.5–1% in aCSF |
| Chemical compound, drug | Texas Red conjugated dextran, 3000 MW, paraformaldehyde, chloral hydrate | Invitrogen, Sigma-Aldrich, Sigma-Aldrich | D3328, 158127, 47335 U | |
| Software, algorithm | ImageJ software | https://imagej.nih.gov/ij/index.html | RRID:SCR_003070 | |
| Software, algorithm | FIJI | | RRID:SCR_002285 | |

*Continued on next page*

*Continued*

| Reagent type (species) or resource | Designation | Source or reference | Identifiers | Additional information |
|---|---|---|---|---|
| Software, algorithm | MetaMorph Basic | Molecular Devices | RRID:SCR_002368 | |
| Software, algorithm | Matlab | Mathworks | RRID:SCR_001622 | |
| Software, algorithm | cellSens | Olympus | RRID:SCR_016238 | |
| Other | artificial CSF | Sigma-Aldrich | | All componets in ACSF (concentrations in mM): 126.0 NaCl, 3 KCl, 2 MgSO$_4$, 10.0 dextrose, 26.0 NaHCO$_3$, 1.25 NaH$_2$PO$_4$, 2 CaCl$_2$ are purchased from Sigma-Aldrich |

## Generation of transgenic mouse lines

### NMU

*Aqp4* KO mice were generated by Dr. Fan Yan by targeted gene disruption as described previously (*Fan et al., 2005*). In brief, an AQP4 replacement targeting vector was constructed using the positive-negative selection cassettes derived from the vectors pPolII long neo bpA and pXhoMC1TK, containing the neoR and HSVtk genes, respectively. The targeting construct was linearized with NotI and introduced into E14K ES cells by electroporation. ES cell clones that were G418/gancyclovir-resistant were isolated, amplified, and screened for targeting fidelity using Southern blot analysis. Cells from two targeted clones were microinjected into CD1 blastocysts and implanted into pseudo-pregnant recipients (*Fan et al., 2005*). Five-month old *Aqp4* KO and WT mice were used in the present study.

### RIKEN

*Aqp4* null/eGFP knock-in mice (SJL.Cg-Aqp4<tm1.1(GFP)Mysi>/MysiRbrc) were generated as described previously (*Ikeshima-Kataoka et al., 2013*). Briefly, knockout of AQP4 was accomplished by replacing the 250 nucleotides in exon 1 (corresponding to the region from Ser 18 to Thr 101) with eGFP cDNA and a PGK-neomycin cassette flanked by flippase recombination target sequences. The targeting vector was electroporated into TT2 ES cells derived from a C57BL/6 female and a CBA male. The PGK-neomycin cassette was removed by crossing with B6-Tg (CAG-FLPe) 37 (*Kanki et al., 2006*). The resultant mice were backcrossed with the SJL/J strain (Charles River, Japan) for more than 12 times (RIKEN CDB accession no. CDB0758K-1; http://www.clst.riken.jp/arg/mutant%20mice%20list.html). Both B6-Tg (CAG-FLPe) 37 and SJL.Cg-Aqp4<tm1.1(GFP)Mysi>/MysiRbrc >mice are provided by the RIKEN BRC through the National Bio-Resource Project of MEXT, Japan (accession no. RBRC01835 and RBRC06500, respectively).

### UNC

The constitutive *Aqp4* KO mouse model was generated by Dr. Alan Verkman (*Ma et al., 1997*). In brief, *Aqp4* KO mice were generated on a CD1 background, following the construction of a targeting vector for homologous recombination of a 7 kb SacI AQP4 genomic fragment, in which part of the exon one coding sequence was deleted. All animal experiments reported in this study were conducted on *Aqp4* KO mice (B6/129), B6/129 controls, and wild-type C57BL/6 mice.

## URMC

*Aqp4* KO mice were generated by Dr. Ole Petter Ottersen using GenOway technique, as described previously (*Thrane et al., 2011*). Their strategy involved cloning and sequencing of a targeted region of the murine *Aqp4* gene in a 129/Sv genetic background. Identification of a targeted locus of the *Aqp4* gene permitted the delete exons 1–3 to avoid any expression of putative splice variants. Hence, a flippase recognition target (FRT)-neomycin-FRT-LoxP–validated cassette was inserted downstream of exon 3, and a LoxP site was inserted upstream of exon 1 (*Thrane et al., 2011*). The mice were backcrossed for 20 + generations with C57BL/6N mice prior to experimentation.

## OHSU

The *Snta1* KO mouse line was generated by Dr. Stanley Froehner (*Adams et al., 2000*), and was purchased through The Jackson Laboratory (*Snta1$^{tm1Scf}$/J, #012940*). In brief, exon 1 of the mouse *Snta1* gene was replaced with a neomycin resistance cassette. The resulting construct was electroporated into 129P2/OlaHsd-derived E14 embryonic, and these cells were then injected into C57BL/6J blastocysts. The resulting mice were bred with C57BL/6J mice for at least 11 generations before the colony was established. Adult male and female mice (WT 15.4 ± 1.6 weeks, *Snta1* KO 14.7 ± 1.6 weeks) were used in these studies.

## Injection, imaging and analysis of CSF tracers or contrast agents
### NMU
#### Intracisternal tracer injections

Intracisternal injections of Texas Red conjugated dextran (TRd3; 3 kDa, Invitrogen) were adapted from a previous report (*Iliff et al., 2012*). In brief, the mice were anesthetized by a 4% chloral hydrate solution. A ten-microliter volume of 0.5% TRd3 dissolved in artificial CSF was infused into the cisterna magna through a 50 µL syringe mounted with a 27-gauge needle (Hamilton, Reno, NV, USA), connected to a constant current syringe pump (TJ-2A/L07-2A, Suzhou Wen Hao Chip Technology Co. Ltd., Jiangsu, China). The intracisternal injection was carried out at a rate of 2 µL/min. Thirty min after the start of injection, the still deeply anesthetized animals were perfused trans-cardially with 4% paraformaldehyde (PFA), and the brains removed and post-fixed in the same fixative for 24 hr.

#### Tissue processing and image analysis of fluorescent tracer

A subset of WT and *Aqp4* KO brains (N = 4 per genotype) were scanned by an ex vivo near infrared (NIR) fluorescence imaging system (Azure biosystems c600, CA, USA). Briefly, the ventral and dorsal brains were imaged (excitation length 660 nm, exposure time: 50 ms), as well as the corresponding bright-field images (*Supplementary file 3*). The fluorescence and bright-field images were superimposed using Adobe Photoshop 8.0. Based on the Mouse Brain Atlas, fluorescence was measured along anterior-posterior axis based on the localization of bregma using NIH Image J software (*Figure 2a*). The mean integrated optical density (MIOD) of TRd3 at a total of 80 sites were sequentially measured from 4 mm anterior to 8 mm posterior to the bregma. In addition, the telencephalon from five mice per genotype was cut into 100 µm coronal sections using a vibratome (Leica, Wetzlar, Germany). Sections were mounted on gelatin-coated slides, washed, and cover-slipped with the slices coated in PBS/glycerol. Images of 6–8 coronal sections collected sequentially from 1.7 mm anterior to 0.7 mm posterior of the bregma were imaged by a Leica DM4000B digital microscope (Leica Microsystems, Wetzlar, Germany) under a 1.25x objective, with uniform exposure time, offset, and gain. The distribution of the tracer (TRd3) was quantified using NIH Image J software as described previously (*Iliff et al., 2012*). In brief, the fluorescent tracer coverage within the whole section was detected using the same threshold and constant settings for minimum and maximum intensities. Subregional analysis of CSF tracer distribution was analyzed in coronal section at the level of 0.5 mm anterior to the bregma. Tracer penetration along perivascular spaces and into brain parenchyma was quantified on high-magnification images of the hypothalamus, which were captured with 10x objective. The fluorescence intensity within the perivascular space was measured along large vessels (diameter >20 µm) extending 500 µm below the brain surface. Two or three separate vessels from each image were analyzed and the data averaged. The fluorescence intensity in the brain parenchyma was also determined along a corresponding linear region adjacent to the vessels (*Figure 2e*).

The imaging and subsequent analysis was performed by an investigator who was blind to animal genotype.

## RIKEN

Adult mice (postnatal 10 to 23 weeks old, mixed gender) were anesthetized by a ketamine-xylazine cocktail (70 mg/kg of ketamine and 10 mg/kg of xylazine) and fixed to a stereotaxic frame. The membrane of the cisterna magna was exposed by surgical procedures. A glass micropipette (tip diameter ~50 µm, preloaded with 0.5% biotinylated dextran amine in saline (BDA, 70 kDa, D1957, Molecular Probes), was carefully pierced through the cisterna magna membrane using a manipulator. A total of 10 µL of the BDA containing solution was infused at a rate of 1 µL/min using a syringe driver (KDS Legato Series, KD Scientific). Thereafter, two supplementary ketamine-xylazine administrations were made (30 min apart, 50% of the initial dose). Five minutes after the second supplementary anesthesia, Mice were transcardially perfused with fixative containing 4% paraformaldehyde in 0.1 M phosphate buffer. After one day of post-fixation in the same fixative, 60 µm slices were incubated in 0.1% triton-X TBS containing Alexa-594-conjugated SA (1:1000) for one hour. Images of SA-processed brain sections were acquired using a standard fluorescent microscope (Olympus BX51 equipped with a DP74 digital camera, *Supplementary file 3*). An offset, defined as the median of the top 1% of pixel intensities of an unprocessed brain section, is subtracted from the raw image of the SA-processed brain section. For depth profiles were calculated from the pial position of 3 mm from the midline using ImageJ.

## UNC
### Intracisternal tracer injections

The experiments were performed as described previously (*Iliff et al., 2012*). In brief, mice (15–18 weeks old) were anesthetized with ketamine/xylazine (100/10 mg/kg) and then a cannula was surgically implanted into the cisterna magna (*Xavier et al., 2018*). Afterwards, an Alexa Fluor 647-conjugated bovine serum albumin tracer was infused into the cistern (0.5% in aCSF; 10 µl at 1 µl/min for 10 min) and allowed to circulate for 30 min. Body temperature was maintained throughout the experiment. Perfusion-fixation is expected to alter intracranial pulsatility, a process that has been shown to drive CSF flow (*Iliff et al., 2013b*); therefore, 30 min post-injection the mice were decapitated, and the brain drop fixed in 4% PFA.

### Tissue processing and microscopy

The mouse brains were post-fixed prior to sectioning. In brief, 100 µm thick coronal plane vibratome sections were obtained using a Leica VT 1200S apparatus (Leica Biosystems, IL). Free-floating mouse brain sections were mounted with ProLong™ Gold Antifade Mounting Media with DAPI (Life Technologies, CA). We used an epifluorescent macroscope for imaging mouse brain sections (Olympus, *Supplementary file 3*). The images were pseudocolored and analyzed on FIJI software. The sections were analyzed following methodology from both *Iliff et al. (2012)* and *Smith et al. (2017)*. Mean pixel intensity was calculated for six coronal sections covering +1.2 to −1.8 mm from anterior to posterior relative to bregma. Subregional analysis of the coronal section at position +0.6 mm was conducted to identify which regions accounted for the differences in CSF influx (*Kress et al., 2014*).

## URMC
### Intracisternal tracer injections

Mice (8- to 10-week-old males) were anesthetized with ketamine/xylazine (100/10 mg/kg) and fixed in a stereotaxic frame whereupon the posterior atlanto-occipital membrane overlying the cisterna magna was surgically exposed. For all CSF tracer experiments, Alexa 647-conjugated bovine serum albumin (BSA-647; 66 kDa; 0.5%) was injected into the subarachnoid CSF via cisterna magna puncture, at a rate of 1 or 2 µL/min for a period of five min (5 µL or 10 µL total volume) through a 30-gauge syringe pump (Harvard Apparatus). To visualize penetration of fluorescent CSF tracers into the brain parenchyma ex vivo, anesthetized animals were decapitated at 30 min after the start of the injection, a time point previously identified to correspond to robust tracer penetration of similar molecular weight compounds in young male C57BL/6 mice (*Iliff et al., 2012*). Brains were then

removed and post-fixed in 4% PFA for 24 hr before being sliced with a vibratome into 100 µm coronal sections which were slide-mounted using PROLONG anti-fade gold with DAPI (Invitrogen).

## In vivo transcranial optical imaging

Transcranial optical imaging was performed in KX-anesthetized mice as recently described (*Plog et al., 2018*). To prepare the mouse, the skin overlying the dorsal convexity was reflected and afterwards the cisterna magna was cannulated as described above. Fluorescent CSF tracers Texas Red 3 kDa dextran and BSA-647 (10 µl at 2 µl/ min, 0.5% in aCSF) were injected into the cisterna magna and imaged using a fluorescent macroscope (MVX10, Olympus) using a PRIOR Lumen 1600-LED light source and ORCA Flash 4.0 digital camera (Hamamatsu). The mouse's head was stabilized with ear bars using a customized stereotaxic frame and placed under the microscope stage. Imaging was done at a 20x magnification and acquired in the far-red emission channel (647 nm). Due to high scattering at the excitation/emission wavelength of the Texas Red 3 kDa dextran through the intact skull this channel was not acquired. Images were collected every minute for 30 min following the start of the injection. MetaMorph Basic imaging software was used for image acquisition (Molecular Devices). The same exposure time (100 ms) was used throughout the experiment and between the WT and KO groups. After 30 min, the brain was extracted and fixed as described below.

## Intrastriatal tracer injections

KX-anesthetized mice were placed in a stereotaxic apparatus and the skin was opened to expose the skull. Carefully, periosteum was removed and bregma was identified with a surgical microscope. A burr hole was done with a hand-held drill on the coordinates AP:+0.5 DV:+3.0 LM:+2.0 mm from bregma, using a 27G needle to break the meningeal layer. A pulled glass capillary (TW100-3, World Precision Instruments) with a 12 µm tip diameter was introduced in the brain parenchyma. Once in place, 20 nL of 0.5% Amyloid-$\beta_{1-40}$ in aCSF (as a sham) was delivered in a perfusion rate of 4 nL/second by a micropump (UMP3 UltraMicroPump with Micro4 controller, World Precision Instruments) as done by Smith et al. Control mice did not receive an injection. The micropipette remained in place for 5 min and then was very slowly withdrew, to avoid any possible backflow. If backflow was observed, the animal was excluded from the experiment. Immediately after the capillary was removed, a small amount of Kwik-Cast Sealant (KWIK-CAST, World Precision Instruments) was applied to seal the site of injection. After 60 min, mice received an injection in cisterna magna with BSA-647 (1% in aCSF) as descried above to evaluate glymphatic function.

## Ex-vivo imaging of fluorescent CSF tracers

BSA-647 and TxRd 3 kDa dextran circulation along perivascular pathways and penetration into the brain parenchyma was visualized by conventional fluorescence microscopy of 100 µm vibratome coronal brain sections, as described previously (*Iliff et al., 2012*). For coronal section quantification, six brain sections (per animal were imaged and analyzed by a blinded investigator using an Olympus fluorescence macroscope (MVX10) under 20x magnification to generate whole-slice images (using the MetaMorph software, Molecular Devices, Supp. File 3). High-magnification montage images of coronal sections were acquired using an Olympus BX51 fluorescent microscope at 4x objective magnification using cellSens software (Olympus). Tracer penetration was quantified using NIH Image J software as described previously (*Iliff et al., 2012*). The first slice was collected at the beginning at the anterior aspect of the corpus callosum, one section was collected every 500 µm apart until a total of six sections had been collected for each animal. Fluorescence of the CSF tracer BSA-647 was measured in each slice as mean pixel intensity (MPI). The MPI from the six brain slices from each animal were averaged to define CSF penetration within a single biological replicate. For the ipsilateral to contralateral comparison, 3 to 5 slices from the injection site where selected from each mouse, and their tracer penetration was quantified, averaged and analyzed by Wilcoxon rank-sum test using different ROIs for each hemisphere. A subset of brain slices were immunostained for AQP4 as previously described (*Kress et al., 2014*). Tracer penetration depth was evaluated using Fiji and Matlab. A coronal section obtained from bregma from each mouse was used for the analysis. In the KO mice, a 500 µm line was placed orthogonal to the cortical surface at the most dorsal position where tracer could be found at the pial surface (*Figure 5f*). Tracer depth in the WT mice was measured at the same position as the KO mouse despite there being more tracer spread over the dorsal convexity. The line profile was normalized to the maximum fluorescence at the cortical surface ($\Delta F/F_{max}$).

Tracer penetration depth was quantified as the distance at which $F_{max}$ was reduced by half ($d_{1/2}$) similar to *Smith et al. (2017)*.

## OHSU

### Dynamic contrast enhanced magnetic resonance imaging (DCE-MRI) of glymphatic transport

Three- to 6-month-old mice were anesthetized with isoflurane, with induction at 3–5%. The posterior atlanto-occipital membrane was exposed surgically. To enable the contrast injection during MRI scanning, we used a pulled glass micropipette with trimmed end (external diameter of approximately. 016 mm) to perform the injection. The micropipette was fixed in place for the duration of the imaging session with superglue. Gadoteridol (68 mM, osmotically adjusted), a contrast medium was infused by syringe pump (Harvard Apparatus) at a rate of 500 nl/min for 20 min (10 µl total volume), with a 2 µl chase of saline.

### MR imaging

CSF circulation was quantified by dynamic contrast-enhanced magnetic resonance imaging (DCE-MRI). All imaging was performed using a Bruker-Biospec 11.75 T preclinical scanner equipped with a 20 mm I.D. quadrature RF volume-coil with a specially designed head holder (*Supplementary file 3*). Heart rate, oxygen saturation and respiratory rate were monitored, and core temperature was maintained at 37°C using a warm air temperature control system (SA Instruments). Upon placement of the glass micropipette, isoflurane anesthesia was switched to ketamine-xylazine (100–10 mg/kg) for the duration of the experiment. Consecutive $T_1$ weighted 3D FLASH images were obtained at 10 min intervals (TR/TE 16/2.8 ms, flip angle 15°, matrix 256 × 192×192, 100 × 100×100 µm resolution), for a total of 90 min. Injection was initiated after acquiring the first image. If no elevation in signal was detectable in the cortical or subcortical brain regions or the basal cistern within the first 30 min, the imaging session was aborted, and the animal was excluded from further analysis.

### MR data analysis

Due to frequent occurrence of motion artifact during the last 30 min of acquisition (70–90 min), these three time points were excluded from analysis, and those animals showing movement artifacts during the first 60 min of acquisition were also excluded. Linear rigid body registration was used to spatially align the image series to the baseline image (FSL). Regions of interest (ROIs) were drawn on a coronal image slice best matching that depicted in *Figure 6I*, and mean intensity values were determined with FIJI software. Repeated measures two-way ANOVA with Sidak multiple comparisons correction was used to analyze the data. Multiple comparisons were used to determine the significance of genotype differences at each time point.

## Meta-analysis

### Literature search

A review of the literature on PubMed and Google Scholar was conducted for all studies published to date (November 2018) on global *Aqp4* KO models in rodents.

### Inclusion criteria:

Out of 6527 studies that were retrieved from the online search, only tracer studies that evaluated fluid flow in the brain in *Aqp4* KO versus wild-type controls were included.

### Exclusion criteria

Studies were excluded if either mean, standard deviation, or sample size could not be extracted. Studies that evaluated tracer transport in a disease model were also eliminated. Non-English studies were not excluded. Only one study Binder et al. 2004 was excluded due to the inability to estimate sample size (*Binder et al., 2004*). After exclusion, a total of 11 studies satisfied the selection criteria. Five data sets from the present study (NMU, RIKEN, UNC, URMC) were also included.

## Data extraction

Means, standard deviations, and sample sizes for each experiment were extracted from the text where possible. If exact numbers were not reported, we used graphical extraction using FIJI. When standard error of the mean was reported, standard deviation was calculated using the square root of the sample size. In cases where sample size was reported as a range, the lowest number was chosen for analysis. In one study, it was impossible to determine if two separate analyses were derived from the same group of mice. In order to avoid inadvertently excluding data (e.g. If the study was in fact conducted in two separate sets of mice) we included both experimental data sets in the meta-analysis (*Smith et al., 2017*).

## Data analysis

Studies addressed mainly two parts of intracranial fluid flow: (1) CSF influx to brain after delivery of tracers to the cisterna magna and (2) tracer transport after being injected into the interstitial fluid in the brain parenchyma. To address these two types of experiments, we decided to run two separate models using the inverse variance method for meta-analysis. Since these were evaluated using different outcome measures between fluorescence (fluorescence intensity and thresholding approaches), radioisotope, and MRI experiments we opted for a standardized mean difference (SMD) using the Cohen's D method for the final analysis. After data was extracted, analysis was done in R (R-Project). Due to high heterogeneity ($I^2 > 70\%$), a random effects model was selected after attempting to reduce heterogeneity by implementing alternate standardized mean difference formulas (e.g. Glass's delta, Hedge's G); however, these attempts did not eliminate heterogeneity. Sources of heterogeneity were then explored using mixed-effects model meta-regression. Candidate predictors of heterogeneity included: anesthesia type, species, strain, age, tracer properties, injection properties, injection location, and experimental method. Inclusion of more than one covariate in the meta-regression resulted in model non-convergence due to the relatively small number of studies and the distribution of these factors across studies so covariates were included one at a time. Data in forest plots is shown as SMD with 95% confidence intervals (CI) and p values less than 0.05 were considered significant.

## Acknowledgement

This study was funded by the NIH (NS100366, NS078394, NS089709, NS061800, AG048769, AG054456, AG054093, NS099371 and HL089221), NNSF81671070, the Knut and Alice Wallenberg Foundation, Western Norway Regional Health Authority (Helse Vest). RIKEN Center for Brain Science (formerly, RIKEN Brain Science Institute), KAKENHI grants (18K14859 to HM, 16H01888 and 18H05150 to HH, 17K19637 and 16H05134 to YA, 18H02606 to MY), HFSP (RGP0036/2014 to HH), JSPS Core-to-Core Program: A Advanced Research Networks, HH is a recipient of the Lundbeck Foundation Visiting Professorship. We thank Dan Xue for expert graphic illustrations, Hayley Martin for assistance with meta-analysis, Amanda M Sweeney and Genaro E Olveda for expert technical assistance, and Paul Cumming for comments on the manuscript.

## Additional information

### Funding

| Funder | Grant reference number | Author |
| --- | --- | --- |
| Japan Society for the Promotion of Science | 18K14859 | Hiromu Monai |
| Knut och Alice Wallenbergs Stiftelse | Helse Vet | Alexander S Thrane |
| Japan Society for the Promotion of Science | 17K19637 | Yoichiro Abe |
| Japan Society for the Promotion of Science | 16H05134 | Yoichiro Abe |

| | | |
|---|---|---|
| Japan Society for the Promotion of Science | 18H02606 | Masato Yasui |
| National Institute on Aging | RF1 AG057575-01 | John H Thomas<br>Maiken Nedergaard |
| Japan Society for the Promotion of Science | 16H01888 | Hajime Hirase |
| Japan Society for the Promotion of Science | 18H05150 | Hajime Hirase |
| Human Frontier Science Program | RGP0036/2014 | Hajime Hirase |
| Japan Society for the Promotion of Science | Core-to-Core Program | Hajime Hirase |
| Lundbeckfonden | Visiting Professorship | Hajime Hirase |
| National Institutes of Health | NS061800 | Aravind Asokan |
| National Institutes of Health | NS099371 | Aravind Asokan |
| National Institutes of Health | HL089221 | Aravind Asokan |
| National Institutes of Health | NS089709 | Jeffrey J Iliff |
| National Institutes of Health | AG054456 | Jeffrey J. Iliff |
| National Institutes of Health | NS100366 | Maiken Nedergaard |
| National Institutes of Health | NS078394 | Maiken Nedergaard |
| National Institutes of Health | AG048769 | Maiken Nedergaard |
| Dr. Miriam and Sheldon G. Adelson Medical Research Foundation | | Maiken Nedergaard |
| EU Joint Programme – Neurodegenerative Disease Research | 643417/DACAPO-AD | Maiken Nedergaard |

The funders had no role in study design, data collection and interpretation, or the decision to submit the work for publication.

## Author contributions

Humberto Mestre, Conceptualization, Data curation, Formal analysis, Investigation, Visualization, Methodology, Writing—original draft, Project administration, Writing—review and editing, Performed meta-analysis; Lauren M Hablitz, Anna LR Xavier, Weixi Feng, Wenyan Zou, Tinglin Pu, Hiromu Monai, Giridhar Murlidharan, Ruth M Castellanos Rivera, Matthew J Simon, Martin M Pike, Virginia Plá, Ting Du, Benjamin T Kress, Conceptualization, Data curation, Formal analysis, Investigation, Visualization, Methodology, Writing—original draft, Writing—review and editing; Xiaowen Wang, Data curation, Investigation, Methodology; Benjamin A Plog, Methodology; Alexander S Thrane, Conceptualization, Visualization, Writing—original draft, Writing—review and editing; Iben Lundgaard, Conceptualization, Writing—original draft, Writing—review and editing; Yoichiro Abe, Masato Yasui, Resources, Funding acquisition, Validation, Investigation, Methodology, Writing—original draft, Writing—review and editing; John H Thomas, Writing—original draft, Writing—review and editing; Ming Xiao, Hajime Hirase, Aravind Asokan, Jeffrey J Iliff, Conceptualization, Resources, Supervision, Funding acquisition, Writing—original draft, Project administration, Writing—review and editing; Maiken Nedergaard, Conceptualization, Resources, Supervision, Funding acquisition, Visualization, Writing—original draft, Project administration, Writing—review and editing

## Author ORCIDs

Humberto Mestre https://orcid.org/0000-0001-5876-5397
Lauren M Hablitz http://orcid.org/0000-0001-6159-7742
Hiromu Monai https://orcid.org/0000-0002-6975-7218
Virginia Plá http://orcid.org/0000-0002-5501-0789

Yoichiro Abe  http://orcid.org/0000-0001-6163-8794

Hajime Hirase  http://orcid.org/0000-0003-3806-6905

Maiken Nedergaard  https://orcid.org/0000-0001-6502-6031

## Ethics

Animal experimentation: All experiments were approved by the Institutional Animal Care and Use Committee of Nanjing Medical University (IACUC-1601106), Wako Animal Experiment Committee, RIKEN (Recombinant DNA experimentation protocol: 2016-038; Animal experimentation protocol: H29-2-227), The University of North Carolina at Chapel Hill Institutional Animal Care and Use Committee (protocol 15-109), the University Committee on Animal Resources of the University of Rochester (protocol 2011-023), and the IACUC of Oregon Health & Science University (protocol IP00000394). All experiments were performed in accordance with the approved guidelines and regulations. All efforts were made to minimize animal suffering and to reduce the number of animals used for the experiments.

## Decision letter and Author response

Decision letter https://doi.org/10.7554/eLife.40070.021

Author response https://doi.org/10.7554/eLife.40070.022

## Additional files

### Supplementary files

• Supplementary file 1. Dataset characteristics from the analysis in *Figure 8a*. All experiments that delivered either fluorescence- or radiolabeled tracers into the cisterna magna of both *Aqp4* KO and wild-type rodents used in the meta-analysis and meta-regression. SD: standard deviation; n: sample size; KO: knockout; KX: ketamine/xylazine; OA: ovalbumin; BDA: biotinylated dextran amine; BSA: bovine serum albumin; N/A: not reported.

DOI: https://doi.org/10.7554/eLife.40070.016

• Supplementary file 2. Dataset characteristics from the analysis in *Figure 8b*. All studies that delivered intracerebral tracers to evaluate clearance or transport of tracers within or out of the brain. SD: standard deviation; n: sample size; KO: knockout; KX: ketamine/xylazine; N/A: not reported; Th: thalamus; Cn: caudate nucleus/striatum; **\*:** value inferred from the text.

DOI: https://doi.org/10.7554/eLife.40070.017

• Supplementary file 3. Imaging parameters.

DOI: https://doi.org/10.7554/eLife.40070.018

• Transparent reporting form

DOI: https://doi.org/10.7554/eLife.40070.019

### Data availability

All data generated or analysed during this study are included in the manuscript.

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
