## [Decision Letter]

Thank you for submitting your article "Aquaporin-4-dependent glymphatic solute transport in the rodent brain" for consideration by *eLife*. Your article has been reviewed by two peer reviewers, and the evaluation has been overseen by a Reviewing Editor and Sean Morrison as the Senior Editor. The reviewers have opted to remain anonymous.

This is a well appreciated consortium effort to address the issue of rebalance of water through the Aquaporin-4 channels. The reviewers have discussed the reviews with one another and the Reviewing Editor has drafted this decision to help you prepare a revised submission.

A number of critical issues must be resolved before a final decision can be made regarding publication.

Data issues

The images of the wild type and APQ4-/- muse shown in Figure 2A, B must be from the same brain areas and must be in good focus and have high dynamic range (you may consider taking the logarithm of the image or using γ ~ 0.5). I would argue that you need to show three to four such pairs as these constitute primary data.

The issue of image quality holds for Figures 4A and 6C-H as well.

In keeping with the current push (and requirement in some journals), please show all data points in Figure 2-5 and 7 as a scatter plot with Tukey boxes, or equivalent rather than means and SDMs; this is similar to what you do in Figure 5.

Please indicate the brain area(s) that are used in each analysis; this could be the range of transverse coordinates relative to bregma.

As the data was taken across many laboratories, please make a table of parameters (brain regions, analysis method, camera, etc.) for each analysis.

A potential physical mechanism should be presented, in the Discussion, for driving CSF from the perivascular to the interstitial and then periventricular spaces, rather than have solution slosh to and fro.

Vocabulary issue:

To avoid overselling, please change title to "Glymphatic solute movement in the rodent brain has an Aquaporin-4-dependent component".

To emphasize the mass action nature of water transfer, please use "flux" rather than "transport", which implies direct energy usage (as in an ATP driven ion exchange).

State explicitly that in this report, all of the data is on influx of CSF as part of the proposed glymphatic system.

Please answer all questions from the specific comments of Reviewer #2.

Reviewer #1:

This manuscript reports evidence from 5 laboratories that deletion of Aquaporin 4 (Aqp4) partially attenuates the movement of labelled solutes injected into the cisterna magna from CSF in the subarachnoid space into the ISF in the peri-vascular space along penetrating arterioles of the mouse cerebral cortex. The main purpose of this particular report is to provide evidence that counters a previous report in *eLife* in which deletion of Aqp4 was found to have no effect on such solute movement. The reason that this matters is that the present authors use this evidence to support a major role for Aqp4 and astroglial endfeet in their hypothetical model of a 'glymphatic' system of active, astrocyte-assisted convection 'transport' of molecules along perivascular spaces for clearance. The authors have proposed (elsewhere) a critical role for this proposed system in clearance of β-amyloid.

In principle, I find the current work sufficiently well done, and the topic of sufficient importance and interest, that the results presented here warrant publication. Nevertheless, I also find that the authors markedly over-state certain aspects of their findings and use terms that are not supported by the findings. For this reason, there are a number of major specific concerns that need to be addressed before moving forward.

1) It is not appropriate to use the term "Aqp4-dependent" in the title, Abstract (second sentence), or anywhere else in the text. The results presented do not demonstrate a "dependence" of solute movement on Aqp4. The most direct and appropriate quantitative evaluation of the effects of Aqp4 deletion in this context appears in Figure 5 and shows an approximate reduction of only 25%. Although this reduction is significant, it is very far from absolute. There is still a majority of detectable solute movement in the absence of Aqp4. Based on this and the other results, the authors can at best say that Aqp4 "facilitates" solute transport, which is the term they use in the final sentence of their paper, or "supports", which is the term they use at the end of the Abstract. The title should not say more than "Aqp4-facilitated", or "Aqp4-supported".

2) As a biologist, I am also not comfortable with the use of the word 'transport' in the manner being done throughout this paper. To me, 'transport' is a term used in biology for something that requires energy and occurs across a barrier, for example transport of molecules across a membrane assisted by a transporter molecule. Here, there is no obvious barrier to the movement of solutes, and there is no obvious requirement for energy to assist what is essentially the passive movement of solutes along a convection flow of fluid current or along a concentration gradient. I would argue that the term 'transport' should also not be used in the title or anywhere in the paper and should be substituted with 'movement' or flux. To me there is a clear distinction between the transport of water across the membrane via Aqp4 and the flow of ISF and the movement of solutes, which to me does not constitute 'transport'. It is noteworthy that the present authors generally use the term 'movement' of solutes throughout the main text. I would be interested to hear other reviewers' comments about this, and of course the authors can provide arguments that they think justify the use of the term 'transport'.

3) NMU data, Figure 2. The results of this experiment seem selectively quantified and over generalized. I am not convinced that the two survey images in 2A show equivalent regions of the brain in the WT and *Aqp4^-/-^* mice. To me it seems likely that the two images shown may be misleading. Both survey images show that the most intense labeling occurs in the subarachnoid space and that this is far more intense along the ventral brain surface. The WT image (left) is clearly taken from a frontal region where there is intensely labeled subarachnoid space between the two hemispheres. In contrast, the *Aqp4^-/-^* image appears to be taken from a region that is relatively similar, but far enough posterior that the hemispheres have already joined and there is no subarachnoid space between the hemispheres. If this is the case, then the comparison is misleading and inappropriately exaggerates the visual appearance of fluorescence intensity in favor of the WT mouse. The images presented are pixelated and blurry, so that the precise location is not readily detectable in the absence of a counterstain such as DAPI. This comparison matters because if the author used these sections for their quantification, that too will have been skewed in favor of the WT mice. The authors need to present images where it is clear to see that they have been take from precisely the same region and where the selective presence of midline subarachnoid tissue does not bias the interpretation in favor of one mouse versus another. In addition, the authors should ensure that this type of bias did not occur in the quantification. The main text states that Figure 2b is taken from hypothalamus, whereas the legend says it is taken from hippocampus. Clearly there is a brain surface involved and so hippocampus seems unlikely. Ideally it would be preferable for the authors to show high magnification images taken from the same sections that are shown in the survey images, and to indicate by using boxes on the survey images where the high magnification images were taken from. Regarding the quantification, the individual data points for each animal should be shown in all of the graphs to represent the scatter. The authors should also indicate which cortical region was used for the perivascular space and parenchymal measurements. Ideally this would correspond to the areas shown in the images. From the results it appears that the measurements were taken only from a ventral brain region. This is somewhat misleading. The survey images that are provided suggest that the lateral and dorsal surfaces of the brain exhibit almost no differences in the penetration of fluorescence from the subarachnoid space. I think other cortical regions should be evaluated from other regions where on the survey images there appears to be no difference in the degree of fluorescence penetration in the WT and *Aqp4^-/-^* mice. If this is not done, then the authors should discuss why there are such large differences in cortical areas and why they feel it is appropriate to select only the region with the largest difference and then generalize their conclusions from this region to the whole brain. Lastly, the authors should make an explicit statement that they ensured that all images used for comparison were collected and evaluated using the same settings for fluorescence intensity and contrast etc.

4) Individual data points must be added to all graphs in all figures. At present it is not possible to judge the scatter and variation of most of the data. This applies to Figures 2, 3, 4 and 6. This information was provided for the present Figure 5 and was also provided for all data in the Smith et al. paper whose work is being challenged. It is a requirement.

Reviewer #2:

General comments:

1) In this group of studies, the authors address intracranial fluid flow including CSF influx to the brain after delivery of tracers to the cisterna magna and tracer transport after injection into the interstitial fluid in the brain parenchyma in four independently generated Aqp4 KO lines as well as an α-syntrophin KO line.

2) Compilation of data from the five independently-conducted experiments showed that deletion or mislocalization of AQP4 suppressed the influx of CSF tracers or contrast agents.

3) The authors interpret these data as supportive evidence for the glymphatic model in which astroglial water transport via AQP4 supports the perivascular influx of cerebral spinal fluid (CSF) and the efflux of interstitial space fluid (ISF).

4) The heterogeneity of the individual studies has to be taken into consideration. The authors do address the concern of conflicting results due to varying methodology used in each study. The paper would be improved if the same methodologies were performed for the independently generated Aqp4 KO lines.

5) Most critically: (1) the authors' hypothesis that anesthetic effects (tribromoethanol vs. ketamine/xylazine) account for the difference between Smith et al. (Smith et al., 2017) results and the current findings is not tested; and (2) no further data are provided in support of the AQP4 dependence of parenchymal clearance of solutes (brain to CSF efflux), as all of the data in the current manuscript relate to CSF to brain penetration (influx).

6) Please see specific comments below.

Specific comments:

1) NMU group injected fluorescent tracer Texas Red-dextran intracisternally which displayed reduced influx in the Aqp4 KO mice as shown in Figure 2 following a thirty-minute infusion. However, this group did not detail location of individual slices collected (6-8 per mouse) as described by the RIKEN group in Figure 3D and UNC group in Figure 4D. These methodological details should be clarified.

2) The authors claim that "High magnification imaging of the hypothalamus revealed that Aqp4 KO impaired the influx…(Figure 2B)" but Figure 2B is neither high magnification nor does it clearly show the hypothalamus. More data would be required to assess this claim.

3) RIKEN and Keio group claim to show that BDA distribution was compromised in Aqp4 KO mice following infusion in Figure 3. However, details of infusion (they say "infusion" in Figure 3A but "injection" in the figure legend) are not clear. Also, was only 1 time point (30 min) performed?

4) Figure 3B is not convincing for a difference between WT and Aqp4 KO.

5) Figure 3C could have been formatted more appropriately to give more clarity (configured similar to Figure 2D). Depth at half-maximum intensity is shown in Figure 3E whereas fluorescence intensity at multiple distances from the brain surface could have provided more information if demonstrated better in Figure 3C. Thus, measurement criteria for penetration depth are heterogeneous across Figures 3C-E.

6) UNC group used the same Aqp4 KO line used in Smith et al. (Smith et al., 2017) in order to assess the reproducibility of the experiments by using methodology reported by Iliff et al. (Iliff et al., 2012). A main justification for disputing the dissimilarity of results is the anesthesia used in the Smith et al. study. The authors argue that the use of tribromoethanol (Avertin) could have effected parenchymal solute efflux yet they didn't test this themselves. It would have been valuable for the UNC group to perform the same experiments using both ketamine/xylazine and tribromoethanol to further support their conclusions. This should have been performed to directly test the idea that the disparity in results is anesthetic state dependent.

7) URMC group claims to show (Figure 5) that Aqp4 deletion reduces entry of Alexa Fluor 647-conjugated BSA from CSF to brain. However, comparison of Figures 5E and 5G and 5F and 5H does not show a convincing difference.

8) In addition, they only looked at mean pixel intensity (MPI) relative to bregma (Figure 5I) and depth was not taken into consideration, unlike the data in other figures in this manuscript. It would have been beneficial to look at MPI respective to definitive depths over a relative distance. This would be a more direct measurement of extent or depth of penetration.

9) OHSU used the Snta1 KO mouse line to access the role of perivascular astrocytic localization of Aqp4 in mediating CSF flux into the parenchyma using DCE-MRI following intracisternal injection of gadoteridol. Figure 6 successfully demonstrates reduced accumulation of Gad in Snta1 KO mice across multiple brain regions.

10) However, Gad is a contrast agent and is not a direct measure of CSF flow itself, thus the conclusion that α-syntrophin facilitates CSF-ISF exchange is not clearly supported.

11) Also, again there is potential confusion created by using both the terms "injection" and "infusion" to describe the experiments (subsection “OHSU: Reduced glymphatic CSF influx in Snta1 KO mice with loss of polarized expression of AQP4 in vascular endfeet of astrocytes”), this should be corrected.

12) All of the experiments done in this paper involve penetration of tracers from CSF injection/infusion into brain parenchyma. None of the experiments directly tests the *clearance* of intraparenchymally infused agents from brain to CSF. This was a prominent finding of Smith et al. (Smith et al., 2017), that there was no significant difference in clearance of solutes in AQP4 KO mice following intraparenchymal infusion. In other words all experiments in this manuscript are CSF to brain, but the claims are bidirectional (Figure 7) that there is also decreased ISF clearance from brain to CSF in the absence of AQP4. Why did none of the current experiments investigate or substantiate this claim?

13) The Discussion reiterates the fact that the UNC group were unable to replicate Smith's findings however they did not follow Smith's methodologies so reproducibility is debatable since methods deviated. This omission reduces the claim of the current manuscript to be a "refutation" of Smith et al.'s findings.

14) An obvious experiment that would eliminate the effects of anesthesia would be to implant an infusion catheter in the cisterna magna and allow animals to recover for several days, and then carry out infusion of tracers in the complete absence of any anesthesia whatsoever and then rapidly perfuse at different time points and assess parenchymal penetration depth criteria. While this may be technically difficult in mice, stereotactically targeted infusions have been performed in awake behaving mice in many locations, and in any event this would be feasible in rats (and AQP4 KO rats have been described). This experiment would clearly move the field forward beyond disparate data relative to anesthetic usage.

---

## [Author Response]

A number of critical issues must be resolved before a final decision can be made regarding publicationData issuesThe images of the wild type and APQ4-/- muse shown in Figure 2A, B must be from the same brain areas and must be in good focus and have high dynamic range (you may consider taking the logarithm of the image or using γ ~ 0.5). I would argue that you need to show three to four such pairs as these constitute primary data.

Following the editor’s suggestions, we have provided in Figure 2, five pairs of the low magnification images of WT and Aqp4 KO forebrain sections that are located at 0.5 mm anterior to the bregma. In addition, we added a new set of analyses based on ex vivo near infrared fluorescence imaging to compare overall tracer (TRd3) distribution along the ventral and dorsal brain regions comparing WT and Aqp4 KO mice. We are now also providing confocal images to obtain higher resolution and eliminate out of focus signal of the coronal brain sections as requested by the reviewer.

The issue of image quality holds for Figures 4A and 6C-H as well.

We have updated Figure 4A by applying a γ of 0.5 per the editor’s suggestion. The images presented in Figure 6C-H are not confocal micrographs, but rather were generated using a 11.75 T preclinical MRI scanner, which is limited in resolution to 100µm^3^ voxels. As such, the images are currently presented at their highest resolution. To clarify this for readers, additional description has been added to the legend of Figure 6.

“Figure 6. Deletion of the adapter protein α-syntrophin impairs AQP4 perivascular localization, and CSF influx into the brain parenchyma. […]Traces for each individual animal are presented (lines) along with the summary statistics (mean ± SEM, 2-way ANOVA). WT n=5, ASYNKO n=7. CTx = cortex (P = 0.0035) Hip = hippocampus (P = 0.0003) Subcortical = subcortical regions (P = 0.0185) 3V = 3^rd^ Ventricle (P = 0.0284) Total (P = 0.0085)”.

In keeping with the current push (and requirement in some journals), please show all data points in Figure 2-5 and 7 as a scatter plot with Tukey boxes, or equivalent rather than means and SDMs; this is similar to what you do in Figure 5.

All data points are now shown for Figures 2-7.

Please indicate the brain area(s) that are used in each analysis; this could be the range of transverse coordinates relative to bregma.As the data was taken across many laboratories, please make a table of parameters (brain regions, analysis method, camera, etc.) for each analysis.

We have included this information in the requested table ‘Imaging Parameters Table’. In addition, we have expanded Figure 1 to include the experimental details from each group.

A potential physical mechanism should be presented, in the Discussion, for driving CSF from the perivascular to the interstitial and then periventricular spaces, rather than have solution slosh to and fro.

We have on request of the editor included a discussion:

“How AQP4 on a cellular level facilitates CSF/ISF exchange remains to be firmly established. […] Moreover, deletion of AQP4 potentiated the increases in intracranial pressure in a murine model of hydrocephalus consistent with the notion that AQP4 supports and facilitates intraparenchymal fluid flow (Ren et al., 2017).”

Vocabulary issue:To avoid overselling, please change title to "Glymphatic solute movement in the rodent brain has an Aquaporin-4-dependent component".

We are puzzled by the objection to our use of the term "Aqp4-dependent." The critique rests on the assertion that we can only use the term if the solute movement depends *only* on Aqp4, which we do not claim. The term “dependent” is commonly used when the dependence is on more than one variable. For example, in the thermodynamics of simple substances, any one thermodynamic variable can be expressed as a function of any two of the other thermodynamic variables, and it is perfectly correct to say, for example, that the density of a gas is “temperature-dependent” when we know that it also depends on the pressure. In the more complicated case of glymphatic flow, the flow no doubt depends on several variables, and we have shown that Aqp4 expression is certainly one of them. The primary purpose of our paper is to refute the claims in the Smith et al., 2017 paper, which claims that solute transport is “aquaporin-4-independent” in its title: therefore, it is highly appropriate that we use the term "Aqp4-dependent" in our title. If we were to use a different term, it would be confusing to the readers.

For clarity, we have in response to the reviewer’s comment added the following sentence to the Discussion:

“However, it is important to note that glymphatic transport is affected by multiple pathways other than AQP4. For example, the sleep-wake cycle (Xie et al., 2013), brain injury in the setting of trauma or ischemia (Iliff et al., 2014; Wang et al., 2017), exercise (He et al., 2017; von Holstein-Rathlou, Petersen and Nedergaard, 2018; Yin et al., 2018), amyloid-β accumulation and acute amyloid-β toxicity (Xu et al., 2015; Peng et al., 2016), omega-3 fatty acids (Ren et al., 2017), plasma osmolarity (Plog et al., 2018), and aging (Kress et al., 2014) are important regulators of glymphatic transport.”

To emphasize the mass action nature of water transfer, please use "flux" rather than "transport", which implies direct energy usage (as in an ATP driven ion exchange).

We respectfully argue that it is preferable to retain the term “transport” in the title of our paper and in the text, with the following justification. We are dealing with bulk flow in an organ (brain) and not carrier or pump mediated transport across a plasma membrane. We use the term precisely as it is used in most papers dealing with solute transport in the brain. Indeed, the primary purpose of our paper is to refute the claims in the Smith et al., 2017 paper, which itself uses the term "transport" in its title ("Test of the 'glymphatic' hypothesis demonstrates diffusive and aquaporin-4-independent solute transport in rodent brain parenchyma") and in the first sentence of its Abstract ("Transport of solutes through brain involves diffusion and convection"). It would be inconsistent and confusing if we were to replace "transport" with a different term in our title. There is a long list of papers that use the term transport in the same context that we have: for example, it is used in the titles of several papers that we cite in our paper (the papers by Asgari et al., Benveniste et al., Holter et al., Murlidharan et al., Plog et al., Ratner et al., and Schley et al.). Also, we point out that the reviewer is incorrect in stating that there is no energy requirement to assist in the movement of solutes along a convective flow: the flow itself requires energy to drive it, in the form of the work done by pressure forces to overcome the resistance due to viscosity.

State explicitly that in this report, all of the data is on influx of CSF as part of the proposed glymphatic system.

We have now included the sentence: “This study focused on re-evaluating the role AQP4 in glymphatic influx of CSF tracers”. In addition, we assessed how intraparenchymal tracer injection affected CSF tracer influx. The latter analysis was included, because the Smith et al., publication refuted the existence of bulk flow based on dispersion of tracers delivered by intraparenchymal injection. We replicated the procedures used by Smith et al. but analyzed its effect of CSF tracer influx. Our analysis showed that the procedures associated with intraparenchymal injection – including preparation of a cranial burr hole, penetration of dura with the glass pipette and tissue injury when the pipette is inserted – consistently suppressed CSF tracer inflow. Based on these observations, we conclude that it is not possible to establish whether convective flow exists in the tissue based on the acute invasive procedures utilized in the Smith et al., 2017 publication.

Please answer all questions from the specific comments of Reviewer #2.Reviewer #1:[…] In principle, I find the current work sufficiently well done, and the topic of sufficient importance and interest, that the results presented here warrant publication. Nevertheless, I also find that the authors markedly over-state certain aspects of their findings and use terms that are not supported by the findings. For this reason, there are a number of major specific concerns that need to be addressed before moving forward.1) It is not appropriate to use the term "Aqp4-dependent" in the title, Abstract (second sentence), or anywhere else in the text. The results presented do not demonstrate a "dependence" of solute movement on Aqp4. The most direct and appropriate quantitative evaluation of the effects of Aqp4 deletion in this context appears in Figure 5 and shows an approximate reduction of only 25%. Although this reduction is significant, it is very far from absolute. There is still a majority of detectable solute movement in the absence of Aqp4. Based on this and the other results, the authors can at best say that Aqp4 "facilitates" solute transport, which is the term they use in the final sentence of their paper, or "supports", which is the term they use at the end of the Abstract. The title should not say more than "Aqp4-facilitated", or "Aqp4-supported".

We are puzzled by the objection to our use of the term "Aqp4-dependent." The critique rests on the assertion that we can only use the term if the solute movement depends *only* on Aqp4, which we do not claim. The term “dependent” is commonly used when the dependence is on more than one variable. For example, in the thermodynamics of simple substances, any one thermodynamic variable can be expressed as a function of any two of the other thermodynamic variables, and it is perfectly correct to say, for example, that the density of a gas is “temperature-dependent” when we know that it also depends on the pressure. In the more complicated case of glymphatic flow, the flow no doubt depends on several variables, and we have shown that Aqp4 concentration is certainly one of them. The primary purpose of our paper is to refute the claims in the Smith et al., 2017 paper, which claims that solute transport is “aquaporin-4-independent” in its title: therefore, it is highly appropriate that we use the term "Aqp4-dependent" in our title. If we were to use a different term, it would be confusing to the readers.

For clarity, we have in response to the reviewer’s comment added the following sentence to the Discussion:

“However, it is important to note that glymphatic transport is affected by multiple pathways other than AQP4. For example, the sleep-wake cycle (Xie et al., 2013), brain injury in the setting of trauma or ischemia (Iliff et al., 2014; Wang et al., 2017), exercise (He et al., 2017; von Holstein-Rathlou, Petersen and Nedergaard, 2018; Yin et al., 2018), amyloid-β accumulation and acute amyloid-β toxicity (Xu et al., 2015; Peng et al., 2016), omega-3 fatty acids (Ren et al., 2017), plasma osmolarity (Plog et al., 2018), and aging (Kress et al., 2014) are important regulators of glymphatic transport.”

2) As a biologist, I am also not comfortable with the use of the word 'transport' in the manner being done throughout this paper. To me, 'transport' is a term used in biology for something that requires energy and occurs across a barrier, for example transport of molecules across a membrane assisted by a transporter molecule. Here, there is no obvious barrier to the movement of solutes, and there is no obvious requirement for energy to assist what is essentially the passive movement of solutes along a convection flow of fluid current or along a concentration gradient. I would argue that the term 'transport' should also not be used in the title or anywhere in the paper and should be substituted with 'movement' or flux. To me there is a clear distinction between the transport of water across the membrane via Aqp4 and the flow of ISF and the movement of solutes, which to me does not constitute 'transport'. It is noteworthy that the present authors generally use the term 'movement' of solutes throughout the main text. I would be interested to hear other reviewers' comments about this, and of course the authors can provide arguments that they think justify the use of the term 'transport'.

We respectfully argue that it is preferable to retain the term “transport” in the title of our paper and in the text, with the following justification. We are dealing with bulk flow in an organ (brain) and not carrier or pump mediated transport across a plasma membrane. We use the term precisely as it is used in most papers dealing with solute transport in the brain. Indeed, the primary purpose of our paper is to refute the claims in the Smith et al., 2017 paper, which itself uses the term "transport" in its title ("Test of the 'glymphatic' hypothesis demonstrates diffusive and aquaporin-4-independent solute transport in rodent brain parenchyma") and in the first sentence of its Abstract ("Transport of solutes through brain involves diffusion and convection"). It would be inconsistent and confusing if we were to replace "transport" with a different term in our title. There is a long list of papers that use the term transport in the same context that we have: for example, it is used in the titles of several papers that we cite in our paper (the papers by Asgari et al., Benveniste et al., Holter et al., Murlidharan et al., Plog et al., Ratner et al., and Schley et al.). Also, we point out that the reviewer is incorrect in stating that there is no energy requirement to assist in the movement of solutes along a convective flow: the flow itself requires energy to drive it, in the form of the work done by pressure forces to overcome the resistance due to viscosity.

3) NMU data, Figure 2. The results of this experiment seem selectively quantified and over generalized. I am not convinced that the two survey images in 2A show equivalent regions of the brain in the WT and Aqp4^-/-^ mice. To me it seems likely that the two images shown may be misleading. Both survey images show that the most intense labeling occurs in the subarachnoid space and that this is far more intense along the ventral brain surface. The WT image (left) is clearly taken from a frontal region where there is intensely labeled subarachnoid space between the two hemispheres. In contrast, the Aqp4^-/-^ image appears to be taken from a region that is relatively similar, but far enough posterior that the hemispheres have already joined and there is no subarachnoid space between the hemispheres. If this is the case, then the comparison is misleading and inappropriately exaggerates the visual appearance of fluorescence intensity in favor of the WT mouse. The images presented are pixelated and blurry, so that the precise location is not readily detectable in the absence of a counterstain such as DAPI. This comparison matters because if the author used these sections for their quantification, that too will have been skewed in favor of the WT mice. The authors need to present images where it is clear to see that they have been take from precisely the same region and where the selective presence of midline subarachnoid tissue does not bias the interpretation in favor of one mouse versus another. In addition, the authors should ensure that this type of bias did not occur in the quantification. The main text states that Figure 2b is taken from hypothalamus, whereas the legend says it is taken from hippocampus. Clearly there is a brain surface involved and so hippocampus seems unlikely. Ideally it would be preferable for the authors to show high magnification images taken from the same sections that are shown in the survey images, and to indicate by using boxes on the survey images where the high magnification images were taken from. Regarding the quantification, the individual data points for each animal should be shown in all of the graphs to represent the scatter. The authors should also indicate which cortical region was used for the perivascular space and parenchymal measurements. Ideally this would correspond to the areas shown in the images. From the results it appears that the measurements were taken only from a ventral brain region. This is somewhat misleading. The survey images that are provided suggest that the lateral and dorsal surfaces of the brain exhibit almost no differences in the penetration of fluorescence from the subarachnoid space. I think other cortical regions should be evaluated from other regions where on the survey images there appears to be no difference in the degree of fluorescence penetration in the WT and Aqp4^-/-^ mice. If this is not done, then the authors should discuss why there are such large differences in cortical areas and why they feel it is appropriate to select only the region with the largest difference and then generalize their conclusions from this region to the whole brain. Lastly, the authors should make an explicit statement that they ensured that all images used for comparison were collected and evaluated using the same settings for fluorescence intensity and contrast etc.

We thank the reviewer very much for raising this critique. We have addressed all issues in the revised manuscript. We are providing the requested detailed information with regard to slice collection and imaging of WT and Aqp4 KO mice in the Materials and methods section. The high magnification images of the hypothalamus have been marked in the low magnification images. We also performed a subregional analysis of CSF tracer in the ventral, lateral and dorsal parts of the forebrain. All data are now displayed as scatter plots. Furthermore, an additional data set has been included. The distribution of CSF tracer (TRd3) at the dorsal and ventral surfaces of the whole-brain is now compared in WT and *Aqp4* KO mice using ex vivo near infrared fluorescence imaging. The corresponding images, quantitative data, and methodological description have been added. To facilitate the review, the new Materials and methods section is as follows:

“Tissue processing and image analysis of fluorescent tracer. A subset of WT and Aqp4KO brains (N = 4 per genotype) were scanned by an ex vivo near infrared (NIR) fluorescence imaging system (Azure biosystems c600, CA, USA). […] The imaging and subsequent analysis was performed by an investigator who was blind to animal genotype.”

4) Individual data points must be added to all graphs in all figures. At present it is not possible to judge the scatter and variation of most of the data. This applies to Figures 2, 3, 4 and 6. This information was provided for the present Figure 5 and was also provided for all data in the Smith et al. paper whose work is being challenged. It is a requirement.

Thanks, individual data points are now displayed in all figures.

Reviewer #2:[…] Specific comments:1) NMU group injected fluorescent tracer Texas Red-dextran intracisternally which displayed reduced influx in the Aqp4 KO mice as shown in Figure 2 following a thirty-minute infusion. However, this group did not detail location of individual slices collected (6-8 per mouse) as described by the RIKEN group in Figure 3D and UNC group in Figure 4D. These methodological details should be clarified.

The methodological details of section collection and imaging analysis have been added to the revised manuscript. The requested data has been added to Figure 2.

2) The authors claim that "High magnification imaging of the hypothalamus revealed that Aqp4 KO impaired the influx…(Figure 2B)" but Figure 2B is neither high magnification nor does it clearly show the hypothalamus. More data would be required to assess this claim.

We have expanded the data analysis presented in Figure 2.

3) RIKEN and Keio group claim to show that BDA distribution was compromised in Aqp4 KO mice following infusion in Figure 3. However, details of infusion (they say "infusion" in Figure 3A but "injection" in the figure legend) are not clear. Also, was only 1 time point (30 min) performed?

We corrected the figure legend to indicate that tracer is introduced by continuous “injection”. We thank the reviewers for this suggestion. In Figure 3A, we now added a blue horizontal bar to indicate the timing and duration of tracer injection. In the respective Materials and methods section, as well as in Figure 1, and in ‘Supplementary file 1’, we have indicated that the CM injection lasted 10 minutes. Yes, the analysis was performed at a single 30 min time point, in line with all the other groups in this submission. This is also the same time point used by both Iliff et al., 2012 and Smith et al., 2017.

4) Figure 3B is not convincing for a difference between WT and Aqp4 KO.

We have changed the dynamic range of the micrographs in Figure 3B to visualize tracer penetration in the parenchyma. The same dynamic range was applied for both WT and *Aqp4^-/-^* images.

5) Figure 3C could have been formatted more appropriately to give more clarity (configured similar to Figure 2D). Depth at half-maximum intensity is shown in Figure 3E whereas fluorescence intensity at multiple distances from the brain surface could have provided more information if demonstrated better in Figure 3C. Thus, measurement criteria for penetration depth are heterogeneous across Figures 3C-E.

We have now reanalyzed and presented our data in a similar style as Figure 2D/E to indicate fluorescence intensity from the brain surface at multiple distances. Indeed, the half distance measure is rather confusing, and we agree that analyzing from the depth 100 µm gives a better insight of tracer penetration difference between WT and *Aqp4^-/-^* mice. For this very reason, we now drop Figures3D and E. A new plot, featuring parenchymal fluorescence signal intensity of layer 2 and below, is displayed as Figure 3D.

6) UNC group used the same Aqp4 KO line used in Smith et al. (Smith et al., 2017) in order to assess the reproducibility of the experiments by using methodology reported by Iliff et al. (Iliff et al., 2012). A main justification for disputing the dissimilarity of results is the anesthesia used in the Smith et al. study. The authors argue that the use of tribromoethanol (Avertin) could have effected parenchymal solute efflux yet they didn't test this themselves. It would have been valuable for the UNC group to perform the same experiments using both ketamine/xylazine and tribromoethanol to further support their conclusions. This should have been performed to directly test the idea that the disparity in results is anesthetic state dependent.

The reviewer is correct in stating that anesthesia is a big factor contributing to the variability. In fact, we have studied in detail the effect of anesthesia on the glymphatic system in a separate submission to another journal. We uploaded the manuscript (Hablitz et al.) as a Related Manuscript File when the manuscript was originally submitted. This manuscript compares glymphatic influx in mice anesthetized with 6 different anesthetic agents. The study included more than 100 mice and documents that ketamine/xylazine is the preferable anesthetics in the study of the glymphatic system. CSF tracer influx under tribromoethanol anesthesia (the one used in Smith et al.) is consistently lower than in ketamine/xylazine anesthesia. We have uploaded the submitted manuscript (Hablitz et al.) again, with the intent that the reviewers have access to it.

7) URMC group claims to show (Figure 5) that Aqp4 deletion reduces entry of Alexa Fluor 647-conjugated BSA from CSF to brain. However, comparison of Figures 5E and 5G and 5F and 5H does not show a convincing difference.

In response to the reviewer’s comments, Figure 5 has been updated. We have included an independent replication of the original experiment. In this experiment, we used a novel in vivo transcranial imaging approach to evaluate CSF tracer influx between wildtypes and *Aqp4* KO mice in Figure 5A and B. We also included a Texas Red 3 kDa dextran tracer in addition to BSA-647 in Figure 5C-E.

8) In addition, they only looked at mean pixel intensity (MPI) relative to bregma (Figure 5I) and depth was not taken into consideration, unlike the data in other figures in this manuscript. It would have been beneficial to look at MPI respective to definitive depths over a relative distance. This would be a more direct measurement of extent or depth of penetration.

We thank the reviewer for this suggestion, we have now included a tracer penetration depth analysis in Figure 5H and I.

9) OHSU used the Snta1 KO mouse line to access the role of perivascular astrocytic localization of Aqp4 in mediating CSF flux into the parenchyma using DCE-MRI following intracisternal injection of gadoteridol. Figure 6 successfully demonstrates reduced accumulation of Gad in Snta1 KO mice across multiple brain regions.10) However, Gad is a contrast agent and is not a direct measure of CSF flow itself, thus the conclusion that α-syntrophin facilitates CSF-ISF exchange is not clearly supported.

Gadoteridol is similar to the fluorescent tracers used in optical imaging studies here employed as a CSF tracer. It is not possible to image water itself and tritiated water moves too quickly across the BBB (seconds) for data collection. Thus, all prior studies are similar to this study based on quantification of tracer dispersion.

11) Also, again there is potential confusion created by using both the terms "injection" and "infusion" to describe the experiments (subsection “OHSU: Reduced glymphatic CSF influx in Snta1 KO mice with loss of polarized expression of AQP4 in vascular endfeet of astrocytes”), this should be corrected.

Thanks, we have in the revised manuscript restricted our use to the term “injection”.

12) All of the experiments done in this paper involve penetration of tracers from CSF injection/infusion into brain parenchyma. None of the experiments directly tests the clearance of intraparenchymally infused agents from brain to CSF. This was a prominent finding of Smith et al. (Smith et al., 2017), that there was no significant difference in clearance of solutes in AQP4 KO mice following intraparenchymal infusion. In other words all experiments in this manuscript are CSF to brain, but the claims are bidirectional (Figure 7) that there is also decreased ISF clearance from brain to CSF in the absence of AQP4. Why did none of the current experiments investigate or substantiate this claim?

We thank the reviewer for this question. We have in the past avoided the use of intraparenchymal injections because the injections (which require opening of the skull and tissue damage) affects CSF tracer influx. Prompted by the reviewer’s question, we have now included a new Figure 7 that formally documents that the invasive methodology utilized by Smith et al., profoundly alters CSF influx and therefore cannot be utilized to study bulk flow clearance in brain.

13) The Discussion reiterates the fact that the UNC group were unable to replicate Smith's findings however they did not follow Smith's methodologies so reproducibility is debatable since methods deviated.

The Smith at al. publication was presented as a replication study: i.e. the title ("Test of the 'glymphatic' hypothesis demonstrates diffusive and aquaporin-4-independent solute transport in rodent brain parenchyma". Please refer to the Discussion of the revised manuscript for a discussion on what a proper replication study is. Our study did not intend to replicate Smith et al., who used (i) an inappropriate anesthetic regiment for study of the glymphatics, (ii) used a novel and untested approach for the cisterna magna injections based on variable injection rate, and (iii) prepared cranial burr holes and introduced glass pipettes to study the glymphatic system. We have, in response to the reviewer, clearly stated that the aim of our study was not to replicate the Smith et al. study. The strength of our study is that it shows that AQP4 is an essential component of the glymphatic system, since 5 independent labs in 5 different transgenic mouse lines and six independent experiments found that deletion or mislocation of AQP4 reduced CSF tracer influx. The data presented in this manuscript is further supported by meta-analysis (Figure 8).

Discussion: “For these reasons, the present study is not a replication study of Smith et al. (Smith et al., 2017). Instead we used the methodology reported in Iliff et al., 2012 (Iliff et al., 2012) and further optimized it in subsequent studies (Neely et al., 2001; Brandt et al., 2014).”

14) An obvious experiment that would eliminate the effects of anesthesia would be to implant an infusion catheter in the cisterna magna and allow animals to recover for several days, and then carry out infusion of tracers in the complete absence of any anesthesia whatsoever and then rapidly perfuse at different time points and assess parenchymal penetration depth criteria. While this may be technically difficult in mice, stereotactically targeted infusions have been performed in awake behaving mice in many locations, and in any event this would be feasible in rats (and AQP4 KO rats have been described). This experiment would clearly move the field forward beyond disparate data relative to anesthetic usage.

We thank the reviewer for this suggestion. The proposed studies are not included here because our aim was to establish the importance of AQP4 in glymphatic activity under anesthesia, not during wakefulness when CSF inflow is very low (Xie et al., 2013). Unfortunately, Dr. Verkman has to our knowledge not yet validated that AQP4 was deleted in the rats utilized in Smith et al., 2017.